# Deep Latent Variable Model based Vertical Federated Learning with Flexible Alignment and Labeling Scenarios

**Kihun Hong    Sejun Park    Ganguk Hwang**[*]
Korea Advanced Institute of Science and Technology (KAIST)
{nuri9911, sejunpark, guhwang}@kaist.ac.kr

## Abstract

Federated learning (FL) has attracted significant attention for enabling collaborative learning without exposing private data. Among the primary variants of FL, vertical federated learning (VFL) addresses feature-partitioned data held by multiple institutions, each holding complementary information for the same set of users. However, existing VFL methods often impose restrictive assumptions such as a small number of participating parties, fully aligned data, or only using labeled data. In this work, we reinterpret alignment gaps in VFL as missing data problems and propose a unified framework that accommodates both training and inference under arbitrary alignment and labeling scenarios, while supporting diverse missingness mechanisms. In the experiments on 168 configurations spanning four benchmark datasets, six training-time missingness patterns, and seven testing-time missingness patterns, our method outperforms all baselines in 160 cases with an average gap of 9.6 percentage points over the next-best competitors. To the best of our knowledge, this is the first VFL framework to jointly handle arbitrary data alignment, unlabeled data, and multi-party collaboration all at once.

## 1 Introduction

With the rapid development of technology, concerns about data privacy and security have been rising, highlighting the importance of research in these areas. Although individuals are reluctant to disclose their personal information for model training, it is indispensable for building more accurate machine learning models. To reconcile these competing interests, federated learning (FL) (McMahan et al., 2017; Zhang et al., 2021) has emerged as a promising paradigm for privacy-preserving model training, leading to the development of numerous methods that address a variety of related challenges.

From a broader perspective, FL can be categorized into three main types: horizontal federated learning (HFL), vertical federated learning (VFL), and federated transfer learning (FTL). Whereas HFL deals with scenarios in which data is distributed across different samples, VFL (Liu et al., 2024; Yang et al., 2023) focuses on cases where data is partitioned by features rather than by samples. For instance, financial institutions and e-commerce platforms, hospitals and pharmaceutical companies, or insurance firms and automobile manufacturers often hold distinct yet complementary information about the same set of users. In such situations where collaborative learning is desired to achieve their respective goals, VFL provides a good framework for training a joint model without directly revealing raw data, thereby preserving privacy.

Building on the VFL paradigm, researchers have explored various strategies to enhance its utility such as improving communication efficiency (Castiglia et al., 2023a;b; Feng, 2022; Liu et al., 2022; Wu et al., 2022), strengthening privacy through advanced security mechanisms (Fu et al., 2022; Kang et al., 2022a; Sun et al., 2024; Zou et al., 2022), and, crucially, boosting the effectiveness (Ganguli et al., 2023; He et al., 2024; Huang et al., 2023; Kang et al., 2022b; Li et al., 2022). A key part of this effort involves fully leveraging available information under restricted data. Unlike conventional machine learning algorithms, VFL typically requires data to be aligned, which is a condition that becomes harder to satisfy as more institutions participate. Although there may be overlap in users

---

[*]Corresponding author

across different institutions, their overall user sets are rarely identical, leaving some users unaligned and even limiting the number of aligned samples. A recent survey on VFL (Wu et al., 2025) reports that only 0.2% of potential VFL pairs are fully alignable. Moreover, in healthcare, banking, or insurance sectors that demand VFL, acquiring labeled data is particularly challenging due to strict privacy regulations and the sensitive nature of the information. Hence, effectively utilizing unaligned or unlabeled data becomes crucial.

Several studies have attempted to address these issues, but frequently rely on restrictive assumptions. Some works are tailored to a small and fixed number of parties (most commonly two) (Kang et al., 2022a;b; Li et al., 2022; Yang et al., 2022), while others do not fully exploit partially aligned data (Feng, 2022; He et al., 2024; Huang et al., 2023), thereby wasting potentially valuable information. In addition, many approaches restrict inference to fully aligned data or specific parties only (Feng, 2022; He et al., 2024; Huang et al., 2023; Kang et al., 2022b), limiting real-world applicability. On the other hand, some researches (Ganguli et al., 2023; Sun et al., 2024) focus on inference with any type of unaligned data but do not consider training on unaligned data.

Before introducing our method, we distinguish two kinds of unaligned data: (a) **Potentially alignable yet currently unlinked records** caused by imperfect identifiers, which can often be addressed via privacy-preserving record-linkage techniques (Hardy et al., 2017; Nock et al., 2021; Wu et al., 2024); and (b) **Inherently unalignable records** that have no genuine counterparts across parties, making alignment fundamentally impossible. Our primary focus is the latter that parallels blockwise missingness in the classical missing data literature. This distinction motivates the following discussion on missingness mechanisms.

Beyond VFL, many studies in machine learning have investigated to tackle missing or incomplete data. A fundamental first step in this line of work is to identify the underlying missingness mechanism, generally categorized into three types (Ghahramani & Jordan, 1995; Little & Rubin, 2019):

- **MCAR** (Missing Completely at Random): Missingness occurs entirely independent of both observed and unobserved values.

- **MAR** (Missing at Random): Missingness may depend on observed values but not on unobserved ones.

- **MNAR** (Missing Not at Random): Missingness may depend on both observed and unobserved values.

Viewing alignment in VFL through the lens of missingness, we can treat unaligned data as in a standard missing data scenario. For example, recent work (Valdeira et al., 2024) applies the MCAR assumption to address arbitrary types of unaligned data, but it does not incorporate unlabeled data. Inspired by prior research (Ipsen et al., 2022), we propose a novel algorithm, FALSE-VFL (**F**lexible **A**lignment and **L**abeling **S**cenarios **E**nabled **V**ertical **F**ederated **L**earning), a unified framework for VFL under diverse alignment and labeling scenarios.

**Contributions.** FALSE-VFL (i) supports both training and inference under arbitrary alignment and labeling conditions in multi-party VFL, (ii) accommodates all three missingness mechanisms (MCAR, MAR, MNAR), and (iii) surpasses all baselines in 160 out of 168 configurations with a substantial performance gap, average of 9.6 percentage points over the next-best competitors. To the best of our knowledge, it is the first framework that simultaneously addresses these challenges in practice.

## 2 RELATED WORK

**Vertical Federated Learning Models.** A variety of VFL approaches have been proposed to handle unaligned or unlabeled data across different institutions. Two-party methods such as VFed-SSD (Li et al., 2022) and FedCVT (Kang et al., 2022b) leverage semi-supervised or psuedo-label training, but cannot scale beyond two parties.

For multi-party settings, subsequent work explores feature selection (VFLFS (Feng, 2022)), representation transfer (VFedTrans (Huang et al., 2023)), self-supervised objectives (FedHSSL (He et al., 2024)), and robustness to dropped parties (MAGS (Ganguli et al., 2023), PlugVFL (Sun et al.,

Table 1: Flexibility of various VFL algorithms.

| VFL Algorithms | Training Data | | | | Unaligned Inference | # Parties |
|---|---|---|---|---|---|---|
| | Labeled | | Unlabeled | | | |
| | Aligned | Unaligned | Aligned | Unaligned | | |
| VFed-SSD (Li et al., 2022) | ✓ | | ✓ | | ✓ | 2 |
| FedCVT (Kang et al., 2022b) | ✓ | ✓ | | ✓ | | 2 |
| VFLFS (Feng, 2022) | ✓ | | | △ | | ≥ 2 |
| VFedTrans (Huang et al., 2023) | | △ | ✓ | | | ≥ 2 |
| MAGS (Ganguli et al., 2023) | ✓ | | | | ✓ | ≥ 2 |
| PlugVFL (Sun et al., 2024) | ✓ | | | | ✓ | ≥ 2 |
| FedHSSL (He et al., 2024) | ✓ | | ✓ | △ | △ | ≥ 2 |
| LASER-VFL (Valdeira et al., 2024) | ✓ | ✓ | | | ✓ | ≥ 2 |
| **FALSE-VFL (Ours)** | ✓ | ✓ | ✓ | ✓ | ✓ | ≥ 2 |

2024)). LASER-VFL (Valdeira et al., 2024) fully exploits aligned and unaligned samples but ignores unlabeled ones.

In Table 1, we summarize the flexibility of these VFL algorithms and our proposed model in terms of data usage, unaligned inference, and the number of parties each method can accommodate. Here, *unaligned inference* refers to whether an algorithm can perform inference on partially and fully unaligned data (see Section 3.1 for the definitions of *partially aligned* and *fully unaligned*). A ✓ indicates that the algorithm fully exploits the corresponding data type, while △ indicates partial exploitation. For example, some approaches may treat partially aligned data as fully unaligned. Also, △ for unaligned inference indicates the algorithm can perform it but require constructing $2^n - 1$ predictors where $n$ is the number of parties. As shown in the table, our proposed model can handle all forms of training data and inference scenarios.

**Deep Latent Variable Models.** Deep latent variable models (DLVMs) (Kingma & Welling, 2014; Rezende et al., 2014) have demonstrated their utility in capturing complex and high-dimensional data structures including datasets with missing values. While these models are widely used in tasks like generative modeling (Burda et al., 2016; Kingma & Welling, 2014; Rezende & Mohamed, 2015), they are also known to be effective in data imputation (Ipsen et al., 2021; Ivanov et al., 2019; Mattei & Frellsen, 2019) which is crucial for handling missing values.

For instance, MIWAE (Mattei & Frellsen, 2019) leverages DLVMs for imputing missing values under the assumption of MAR mechanism, while not-MIWAE (Ipsen et al., 2021) extends it to handle MNAR scenarios. Furthermore, supMIWAE (Ipsen et al., 2022) introduces a supervised learning framework for incomplete datasets, imputing missing values under MAR assumption.

These advances in DLVMs illustrate the flexibility and power of probabilistic models in addressing missing data challenges, enabling effective imputation strategies depending on the nature of the missingness mechanism.

## 3 OUR METHOD: FALSE-VFL

### 3.1 DATA SETTING AND NOTATIONS

We consider a vertical federated learning (VFL) setting with one active party which possesses labels and $K - 1$ passive parties. Each party $k \in [K] := \{1, 2, \cdots, K\}$ holds a set of observations $\{\boldsymbol{x}_i^k \in \mathbb{R}^{d_k}\}_{i=1}^N$. For each sample $i$, the complete observation across all parties is denoted by $\boldsymbol{x}_i := [\boldsymbol{x}_i^1, \boldsymbol{x}_i^2, \cdots, \boldsymbol{x}_i^K]$. Only the active party $K$ owns the labels $\{y_i \in \mathbb{R}\}_{i=1}^N$ which correspond to the complete observations $\{\boldsymbol{x}_i \in \mathbb{R}^d\}_{i=1}^N$ where $d = \sum_{k=1}^K d_k$. Critically, both the observations and labels are private, meaning that they cannot be shared directly among the parties.

In real-world scenarios, data may be incomplete with missing labels or unaligned observations across parties. We address this by introducing the following assumptions regarding data availability. Let $m_i^k \in \{0, 1\}$ indicate whether the data $\boldsymbol{x}_i^k$ is observed in party $k$, i.e., $m_i^k = 0$ denotes it is observed

and $m_i^k = 1$ denotes it is not. Define $\boldsymbol{m}_i := [m_i^1, m_i^2, \cdots, m_i^K]$ to represent the availability of observations across all parties. Similarly, let $u_i \in \{0, 1\}$ indicate whether the label $y_i$ is available with $u_i = 1$ representing a missing label.

Each sample $\boldsymbol{x}_i$ can be split into $[\boldsymbol{x}_i^{obs}, \boldsymbol{x}_i^{mis}]$ where $\boldsymbol{x}_i^{obs}$ denotes the observed parts (i.e., the set of $\boldsymbol{x}_i^k$ where $m_i^k = 0$) and $\boldsymbol{x}_i^{mis}$ denotes the missing parts. Importantly, we only have access to $\boldsymbol{x}_i^{obs}$ and the missing data $\boldsymbol{x}_i^{mis}$ is not available. Additionally, if $u_i = 1$, the corresponding label $y_i = \text{NA}$. Under this framework, the set $\{\boldsymbol{x}_i^{obs} \mid u_i = 1, i \in [N]\}$ represents the unlabeled data, while $\{\boldsymbol{x}_i^{obs} \mid u_i = 0, i \in [N]\}$ represents the labeled data.

The alignment of observations across parties can be represented with our missingness notation. **Fully aligned** observations where all parties contribute to the data, are represented as $\{\boldsymbol{x}_i \mid \sum_{k=1}^K m_i^k = 0, i \in [N]\}$. **Fully unaligned** observations where data from all but one party is missing, are represented as $\{\boldsymbol{x}_i^{obs} \mid \sum_{k=1}^K m_i^k = K - 1, i \in [N]\}$. The remaining observations where $0 < \sum_{k=1}^K m_i^k < K - 1$, are considered **partially aligned**. For simplicity, we refer to fully aligned observations as "aligned" and consider both fully unaligned and partially aligned observations as "unaligned".

This formulation provides a unified framework for addressing incomplete data, accommodating both unlabeled and unaligned observations.

## 3.2 Problem and Approach

In this work our goal is to solve supervised learning tasks including regression or classification tasks with deep neural architectures on the training dataset $\{\boldsymbol{x}_i^{obs}, y_i, \boldsymbol{m}_i, u_i\}_{i=1}^N$ including unlabeled data and incomplete observations. If complete observations were available, the standard approach would involve maximizing the log-likelihood function $\sum_{u_i=0, i \in [N]} \log p_\Theta(y_i|\boldsymbol{x}_i)$. However, due to the significant presence of unlabeled data and incomplete observations, more sophisticated procedures are required.

Inspired by the methodology in Ipsen et al. (2022), we leverage deep latent variable models (DLVMs) (Burda et al., 2016; Kingma & Welling, 2014; Rezende et al., 2014) to build a new predictive model $p_{\Theta,\psi}(y|\boldsymbol{x}^{obs}, \boldsymbol{m})$ where $\psi$ parameterizes the function between $\boldsymbol{x}$ and $\boldsymbol{m}$, while incorporating unlabeled data into the training process. Under the MAR assumption, we have $p_{\Theta,\psi}(y|\boldsymbol{x}^{obs}, \boldsymbol{m}) = p_\Theta(y|\boldsymbol{x}^{obs})$ as shown in Appendix A.1. We therefore present two version of our method:

- **FALSE-VFL-I**: assumes MAR mechanism and is detailed in Sections 3.3 to 3.5;
- **FALSE-VFL-II**: relaxes assumption MAR to MNAR mechanism and is described in Appendix A.2.

Below we outline how the approach of Ipsen et al. (2022) is integrated into FALSE-VFL-I (MAR case).

To effectively utilize the large amount of unlabeled data, as a pretraining step we first consider the marginal log-likelihood $\sum_{i \in [N]} \log p_{\Theta_g}(\boldsymbol{x}_i^{obs})$ and maximize it where $\Theta_g \subset \Theta$ represents the generative model parameters. To model the generation of observations, we introduce a sequence of stochastic hidden layers with latent variables $\boldsymbol{h} = \{\boldsymbol{h}^1, \cdots, \boldsymbol{h}^L\}$ as described in Burda et al. (2016):

$$p_{\Theta_g}(\boldsymbol{x}) = \int p_{\Theta_g}(\boldsymbol{h}^L) p_{\Theta_g}(\boldsymbol{h}^{L-1}|\boldsymbol{h}^L) \cdots p_{\Theta_g}(\boldsymbol{x}|\boldsymbol{h}^1) \, d\boldsymbol{h}.$$

Next, as a training step we leverage the labeled data to maximize the log-likelihood function $\sum_{u_i=0, i \in [N]} \log p_\Theta(y_i|\boldsymbol{x}_i^{obs})$. Since raw observations $\boldsymbol{x}_i^{obs}$ cannot be shared across parties due to privacy constraint, we rely on the latent variable $\boldsymbol{h}^1$, the output of the stochastic layer preceding the observations, to generate the corresponding labels. This structure results in a graphical model illustrated in Fig. 1 where $\boldsymbol{h} := \boldsymbol{h}^1$ and $\boldsymbol{z} := \boldsymbol{h}^L$.[1] The detailed explanation of Fig. 1 will be provided in Section 3.3. Note that, for reasons of computational tractability, we fix the parameters $\Theta_g$ after

---

[1]In the whole remaining context, we use $L = 2$ for simplicity. However, note that we can use any $L \geq 2$ with a small effort.

pretraining and maximize the conditional log-likelihood $\sum_{u_i=0, i\in[N]} \log p_\Theta(y_i|\boldsymbol{x}_i^{obs})$ with fixed $\Theta_g$, which is shown to be equivalent to maximizing the joint log-likelihood $\sum_{u_i=0, i\in[N]} \log p_\Theta(y_i, \boldsymbol{x}_i^{obs})$. We will explain how the pretraining and training steps can achieve our goal effectively in Section 4.

### 3.3 MODEL ARCHITECTURE

From a broader perspective, our model is divided into feature-side and label-side modules. Each party $k \in [K]$ has its own encoder (parameterized by $\gamma_c^k$) and decoder (parameterized by $\theta_c^k$), which we collectively refer to as the feature-side modules. Meanwhile, the active party which holds the labels additionally has a global encoder (parameterized by $\gamma_s$), a global decoder (parameterized by $\theta_s$), and a discriminator (parameterized by $\phi$) which form the label-side modules. For simplicity, we define the collections of feature-side parameters as $\gamma_c = \{\gamma_c^1, \cdots, \gamma_c^K\}$ and $\theta_c = \{\theta_c^1, \cdots, \theta_c^K\}$. Thus, we can express the overall model parameters $\Theta$ and $\Theta_g$ as

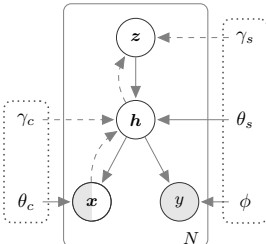

Figure 1: Graphical model for FALSE-VFL-I.

$$\Theta = \{\gamma_c, \theta_c, \gamma_s, \theta_s, \phi\} \text{ and } \Theta_g = \{\gamma_c, \theta_c, \gamma_s, \theta_s\}.$$

All encoders and decoders produce parameters for pre-defined distributions (e.g., mean and variance for Gaussian distributions). Specifically, we assume that the prior distribution $p(\boldsymbol{z})$ is a standard Gaussian, while $p_{\theta_s}(\boldsymbol{h}|\boldsymbol{z})$ and $p_{\theta_c}(\boldsymbol{x}|\boldsymbol{h}) = \prod_{k\in[K]} p_{\theta_c^k}(\boldsymbol{x}^k|\boldsymbol{h})$ are modeled as Gaussian distributions.

We employ amortized variational inference to approximate the posterior distributions. Specifically, the approximate posterior $q_{\gamma_c}(\boldsymbol{h}|\boldsymbol{x}^{obs})$ for the latent variable $\boldsymbol{h}$ is given by Gaussian distribution

$$\mathcal{N}\left(\frac{1}{|obs|}\sum_{k\in obs}\boldsymbol{\mu}_{\gamma_c^k}(\boldsymbol{x}^k), \left(\sum_{k\in obs}\boldsymbol{\Sigma}_{\gamma_c^k}^{-1}(\boldsymbol{x}^k)\right)^{-1}\right), \quad (1)$$

where $\boldsymbol{\mu}_{\gamma_c^k}(\boldsymbol{x}^k)$ and $\boldsymbol{\Sigma}_{\gamma_c^k}(\boldsymbol{x}^k)$ denote the outputs of encoder from party $k$. The approximate posterior $q_{\gamma_s}(\boldsymbol{z}|\boldsymbol{h})$ for $\boldsymbol{z}$ is also assumed to be Gaussian. The rationale behind the formulation in (1) will be explained in Appendix B. For the discriminative part, we assume $p_\phi(y|\boldsymbol{h})$ to be a Gaussian distribution for regression tasks and to be a categorical distribution for classification tasks. The complete computational flow is summarized in Fig. 2. With this setup, we are now ready to explain the whole algorithm including pretraining and prediction.

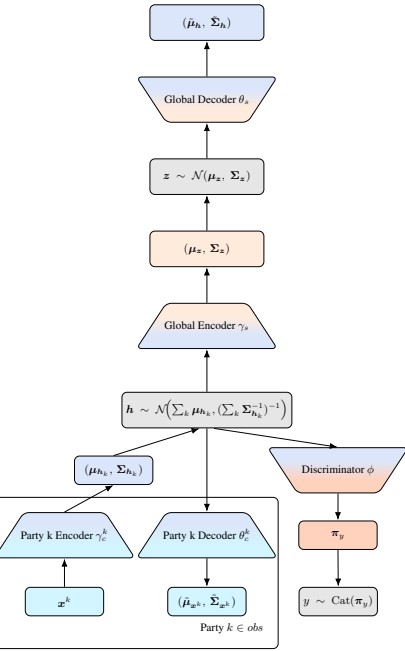

Figure 2: Computational structure for FALSE-VFL-I.

### 3.4 ALGORITHM

Our method consists of three steps: pretraining, training, and prediction. We present the details of our algorithm below and provide its overview in Algorithm 1 of Appendix D.3.

**Pretraining with Marginal Likelihood Maximization.** From our graphical model assumption, the log-likelihood of the observed features can be written as:

$$\log p_{\Theta_g}(\boldsymbol{x}^{obs}) = \log \int p_{\theta_c}(\boldsymbol{x}^{obs}|\boldsymbol{h})p_{\theta_s}(\boldsymbol{h}|\boldsymbol{z})p(\boldsymbol{z}) \, d\boldsymbol{h}d\boldsymbol{z}.$$

This integral is intractable, so we approximate it using $\kappa$-sample importance-weighted estimator (Burda et al., 2016) with the approximate posterior. First, define

$$R_\kappa(\boldsymbol{x}^{obs}) := \frac{1}{\kappa}\sum_{j=1}^{\kappa} \frac{p_{\theta_c}(\boldsymbol{x}^{obs}|\boldsymbol{h}_j)p_{\theta_s}(\boldsymbol{h}_j|\boldsymbol{z}_j)p(\boldsymbol{z}_j)}{q_{\gamma_c}(\boldsymbol{h}_j|\boldsymbol{x}^{obs})q_{\gamma_s}(\boldsymbol{z}_j|\boldsymbol{h}_j)}.$$

where $\{(\boldsymbol{h}_j, \boldsymbol{z}_j)\}$ are i.i.d. random variables with distribution $q_{\gamma_c}(\boldsymbol{h}|\boldsymbol{x}^{obs})q_{\gamma_s}(\boldsymbol{z}|\boldsymbol{h})$. By construction, $R_\kappa(\boldsymbol{x}^{obs})$ is an unbiased estimator of $p_{\Theta_g}(\boldsymbol{x}^{obs})$. Applying Jensen's inequality to $\log \mathbb{E}[R_\kappa(\boldsymbol{x}^{obs})]$, we get the lower bound $\mathcal{L}_\kappa(\Theta_g)$ of the log-likelihood of the observed features:

$$\sum_{i\in[N]} \log p_{\Theta_g}(\boldsymbol{x}_i^{obs}) \geq \mathcal{L}_\kappa(\Theta_g) := \sum_{i\in[N]} \mathbb{E}_{\{(\boldsymbol{h}_j,\boldsymbol{z}_j)\}_{j=1}^\kappa \sim q_{\gamma_c}(\boldsymbol{h}|\boldsymbol{x}_i^{obs})q_{\gamma_s}(\boldsymbol{z}|\boldsymbol{h})} \left[\log R_\kappa(\boldsymbol{x}_i^{obs})\right].$$

Finally, we just maximize $\mathcal{L}_\kappa$ with respect to $\Theta_g$.

**Training with Conditional Likelihood Maximization.** After we pretrain our model with marginal likelihood maximization, we fix the parameters $\Theta_g$. Similarly as above, let

$$R'_\kappa(y, \boldsymbol{x}^{obs}) = \frac{1}{\kappa} \sum_{j=1}^\kappa \frac{p_\phi(y|\boldsymbol{h}_j)p_{\theta_c}(\boldsymbol{x}^{obs}|\boldsymbol{h}_j)p_{\theta_s}(\boldsymbol{h}_j|\boldsymbol{z}_j)p(\boldsymbol{z}_j)}{q_{\gamma_c}(\boldsymbol{h}_j|\boldsymbol{x}^{obs})q_{\gamma_s}(\boldsymbol{z}_j|\boldsymbol{h}_j)}.$$

Then, we also get the lower bound $\mathcal{L}'_\kappa(\phi)$ of the log-likelihood of the labeled data:

$$\sum_{u_i=0, i\in[N]} \log p_\Theta(y_i, \boldsymbol{x}_i^{obs}) \geq \mathcal{L}'_\kappa(\phi) := \sum_{\substack{u_i=0, \\ i\in[N]}} \mathbb{E}_{\{(\boldsymbol{h}_j,\boldsymbol{z}_j)\}_{j=1}^\kappa \sim q_{\gamma_c}(\boldsymbol{h}|\boldsymbol{x}_i^{obs})q_{\gamma_s}(\boldsymbol{z}|\boldsymbol{h})} \left[\log R'_\kappa(y_i, \boldsymbol{x}_i^{obs})\right].$$

Finally, we maximize $\mathcal{L}'_\kappa$ with respect to $\phi$.

**Prediction.** After all training steps are done, we can predict the label of new (incomplete) observations with self-normalized importance sampling method as in Ipsen et al. (2022):

$$p_\Theta(y|\boldsymbol{x}^{obs}) \approx \sum_{l=1}^L w_l p_\phi(y|\boldsymbol{h}_l), \quad \text{where} \quad w_l = \frac{r_l}{r_1 + \cdots + r_L}, \quad r_l = \frac{p_{\theta_c}(\boldsymbol{x}^{obs}|\boldsymbol{h}_l)p_{\theta_s}(\boldsymbol{h}_l|\boldsymbol{z}_l)p(\boldsymbol{z}_l)}{q_{\gamma_c}(\boldsymbol{h}_l|\boldsymbol{x}^{obs})q_{\gamma_s}(\boldsymbol{z}_l|\boldsymbol{h}_l)},$$

and $(\boldsymbol{h}_1, \boldsymbol{z}_1), \cdots, (\boldsymbol{h}_L, \boldsymbol{z}_L)$ are i.i.d. samples from $q_{\gamma_c}(\boldsymbol{h}|\boldsymbol{x}^{obs})q_{\gamma_s}(\boldsymbol{z}|\boldsymbol{h})$.

**Convergence Properties.** We summarize here convergence properties of $\mathcal{L}_\kappa$ and $\mathcal{L}'_\kappa$. The formal statement with the mild regularity conditions and the proof are given in Appendix C.

**Theorem 3.1.** $\mathcal{L}_\kappa \left(\mathcal{L}'_\kappa, resp.\right)$ *increases as* $\kappa$ *increases, and bounded above by* $\log p(\boldsymbol{x}^{obs})$ $\left(\log p(y, \boldsymbol{x}^{obs}), resp.\right)$. *In addition,* $\mathcal{L}_\kappa \left(\mathcal{L}'_\kappa, resp.\right)$ *converges to* $\log p(\boldsymbol{x}^{obs})$ $\left(\log p(y, \boldsymbol{x}^{obs}), resp.\right)$ *as* $k \to \infty$ *under mild regularity conditions.*

### 3.5 COMMUNICATION BETWEEN PARTIES

In the pretraining and training steps, each party computes the mean and variance of its approximate posterior and sends this local latent representation to the active party. The active party aggregates the local latent representations into a global latent distribution, samples latent variables $\boldsymbol{h}$ from this distribution, and broadcasts them to the participating parties so that they can evaluate $p_{\theta_c}(\boldsymbol{x}^{obs}|\boldsymbol{h})$ via their local decoders. Each passive party returns the scalar probabilities to the active party which uses them to compute the loss and sends back gradients for the local model parameters of each party.

In the inference step, we follow the same forward communication pattern as in the pretraining and training phases, but no gradients are exchanged.

Thus, compared to the standard VFL, FALSE-VFL-I follows the same basic communication pattern where local representations are sent from the passive parties to the active party and the gradients are sent in the reverse direction, but with two additional steps: sampled latent variables from the global posterior sent from the active party to the others, and scalar probabilities sent back to the active party.

## 4 MAXIMIZING CONDITIONAL LIKELIHOOD THROUGH TWO-STAGE TRAINING

The primary objective of our work is to predict target variables $y$ (either continuous or discrete) based on observed features $\boldsymbol{x}^{obs}$. To achieve it, we aim to maximize the conditional log-likelihood

$\sum_{u_i=0, i \in [N]} \log p_\Theta(y_i | \boldsymbol{x}_i^{obs})$. However, this objective is intractable even with variational approximation. To address this issue, we propose a detour by maximizing the joint log-likelihood $\sum_{u_i=0, i \in [N]} \log p_\Theta(y_i, \boldsymbol{x}_i^{obs})$ which can be optimized using variational approximations.

The connection between the joint and conditional log-likelihoods is expressed as:

$$\log p_\Theta(y, \boldsymbol{x}^{obs}) = \log p_\Theta(y | \boldsymbol{x}^{obs}) + \log p_{\Theta_g}(\boldsymbol{x}^{obs})$$

This relationship indicates that maximizing the joint likelihood $p_\Theta(y, \boldsymbol{x}^{obs})$ inherently involves both the conditional term $p_\Theta(y | \boldsymbol{x}^{obs})$ and the marginal term $p_{\Theta_g}(\boldsymbol{x}^{obs})$. Due to the high dimensionality of $\boldsymbol{x}^{obs}$ where $d^{obs} \gg 1$, the term $p_{\Theta_g}(\boldsymbol{x}^{obs})$ may dominate the learning process, leading to implicit modeling bias as discussed in Zhao et al. (2019). This imbalance can result in a model focusing more on the marginal likelihood and less on the conditional likelihood, ultimately limiting the performace of $p_\Theta(y | \boldsymbol{x}^{obs})$.

**Two-Stage Training Strategy**   To mitigate this imbalance, we introduce a two-stage training process:

- Stage 1 - Pretraining with Marginal Likelihood Maximization: To avoid the implicit modeling bias, we first maximize the marginal likelihood $p_{\Theta_g}(\boldsymbol{x}^{obs})$ which constitutes the dominant part of the joint likelihood $p_\Theta(y, \boldsymbol{x}^{obs})$.
- Stage 2 - Training with Conditional Likelihood Maximization: Once $p_{\Theta_g}(\boldsymbol{x}^{obs})$ is optimized, we freeze the parameters $\Theta_g$ and proceed to maximize the joint likelihood $p_\Theta(y, \boldsymbol{x}^{obs})$. Since $\Theta_g$ is fixed, this step effectively focuses on maximizing the conditional likelihood $p_\Theta(y | \boldsymbol{x}^{obs})$, aligning directly with our goal of optimizing conditional predictions.

Freezing $\Theta_g$ not only reduces the implicit modeling bias from $p_{\Theta_g}(\boldsymbol{x}^{obs})$, but also makes the objective of maximizing $p_\Theta(y, \boldsymbol{x}^{obs})$ equivalent to maximizing $p_\Theta(y | \boldsymbol{x}^{obs})$. This two-stage approach provides a more focused path towards optimizing the conditional likelihood. Moreover, it also has the following important properties.

**Interpretation as Feature Learning**   The two-stage training process can be also interpreted in terms of feature learning. In this view, the first step (maximizing $p_{\Theta_g}(\boldsymbol{x}^{obs})$) serves as a pretraining phase where the generative model learns a robust representation of the feature space. Once this representation is well-learned, the second step leverages this generative model to facilitate the maximization of $p_\Theta(y | \boldsymbol{x}^{obs})$, improving the conditional prediction.

**Incorporating Unlabeled Data**   An additional advantage of this two-stage approach is the ability to incorporate unlabeled data. Given the abundance of unlabeled data in many practical scenarios, optimizing $p_{\Theta_g}(\boldsymbol{x}^{obs})$ allows the model to utilize them, even though labels are absent. While unlabeled data cannot directly optimize $p_\Theta(y | \boldsymbol{x}^{obs})$, it plays a crucial role in optimizing the marginal likelihood $p_{\Theta_g}(\boldsymbol{x}^{obs})$, thus ensuring the model effectively leverages all available data during training.

## 5   EXPERIMENTS

In this section, we compare FALSE-VFL with several baseline algorithms on four benchmark datasets.

**Baselines.**   Vanilla VFL, LASER-VFL (Valdeira et al., 2024), PlugVFL (Sun et al., 2024), and FedHSSL (He et al., 2024) are evaluated; implementation specifics for each baseline are provided in Appendix D.1.

**Datasets and Models.**   We use Isolet (Cole et al., 1990), HAPT (Reyes-Ortiz et al., 2016), FashionMNIST (Xiao, 2017), and ModelNet10 (Wu et al., 2015). Isolet and HAPT are tabular, whereas FashionMNIST and ModelNet10 are image datasets. For tabular tasks we employ simple multilayer perceptrons, while for image tasks we adopt ResNet-18 backbone (He et al., 2016) as the feature extractors and two fully-connected layers for the fusion models. More detailed explanations on datasets and models appear in Appendix D.2.

**Setup.** Isolet, HAPT, and FashionMNIST are partitioned across eight parties, and ModelNet10 across six; exact feature splits are explained in Appendix D.2. We provide 500 labeled samples for each tabular dataset and 1,000 for each image dataset, treating all remaining samples as unlabeled. Since Vanilla VFL, PlugVFL, and FedHSSL originally require fully aligned data during training, we reserve 100 (tabular) or 200 (image) of the labeled samples as fully aligned and render the remainder partially aligned according to the designated train data missing mechanisms. All test samples are partially aligned, using the designated test data missing mechanisms. Section 5.1 explains the missing mechanisms we adopt.

## 5.1 MISSING MECHANISMS

We evaluate all algorithms under MCAR, MAR, and MNAR assumptions for data alignment.

**MCAR:** To generate unaligned data with the MCAR mechanism, we impose a missing probability $p$ for each sample in each party. We experiment with $p \in \{0.0, 0.2, 0.5\}$ and denote as MCAR 0, MCAR 2, and MCAR 5, respectively.

**MAR:** We designs two MAR mechanisms, MAR 1 and MAR 2, motivated by real-world scenarios in which subsequent observations depend on what has already been seen. MAR 1 is designed to seek for single highly informative party, whereas MAR 2 is designed to seek multiple moderately informative parties. Precise formulas are given in Appendix D.4.

**MNAR:** After normalizing each dataset, let $\bar{x}^k$ be the mean of features held by party $k$ in a given sample. If $\bar{x}^k < 0$, the party is dropped with probability $p$; otherwise it is dropped with probability $1 - p$. We experiment with $p \in \{0.7, 0.9\}$ and denote as MNAR 7 and MNAR 9, respectively.

## 5.2 EXPERIMENTAL RESULTS

We train every method on the four benchmarks under six training missingness regimes (MCAR 2, MCAR 5, MAR 1, MAR 2, MNAR 7, MNAR 9) and evaluate them on seven test patterns (MCAR 0, MCAR 2, MCAR 5, MAR 1, MAR 2, MNAR 7, MNAR 9). The results are shown in Fig. 3, and exact numerical values are reported in Tables 13 to 18. Although FALSE-VFL-I does not model the mask distribution explicitly, its performance is consistently close to, and occasionally higher than, that of FALSE-VFL-II. To provide a concise overview, we compare the best-performing FALSE-VFL variant with the strongest baseline for each dataset, training mechanism, and test mechanism in Table 2. As shown in Table 2c, the performace gap widens as the test missingness rate increases. In all comparisons, FALSE-VFL achieves a clear lead.

Table 2: Average accuracy gap (%) between the **best FALSE-VFL variant** and the **best competing baseline**.

(a) Per dataset

|  | Isolet | HAPT | F-MNIST | ModelNet |
|---|---|---|---|---|
| Gap | 15.7 | 7.3 | 4.5 | 9.1 |

(b) Per *training* missing data mechanism

|  | MCAR 2 | MCAR 5 | MAR 1 | MAR 2 | MNAR 7 | MNAR 9 |
|---|---|---|---|---|---|---|
| Gap | 5.3 | 10.4 | 9.2 | 9.6 | 11.4 | 8.9 |

(c) Per *test* missing data mechanism

|  | MCAR 0 | MCAR 2 | MCAR 5 | MAR 1 | MAR 2 | MNAR 7 | MNAR 9 |
|---|---|---|---|---|---|---|---|
| Gap | 1.8 | 5.4 | 9.6 | 10.9 | 12.3 | 10.8 | 13.1 |

Overall, FALSE-VFL exceeds the strongest baseline by an average of 9.1 percentage points across all 168 configurations. This average differs slightly from 9.6 reported in the abstract since the value is calculated on 160 cases.

**Robustness to higher missing rates.** To assess the impact of more severe missingness, we compare models trained under MCAR 5 with those trained under MCAR 2. Table 19 reports, for each setting, the ratio of mean accuracy obtained when training under MCAR 5 to that obtained under MCAR 2. FALSE-VFL exhibits the smallest performance decrease in 23 of 28 cases and even improves on most test sets, demonstrating markedly stronger robustness to missingness than the competing methods.

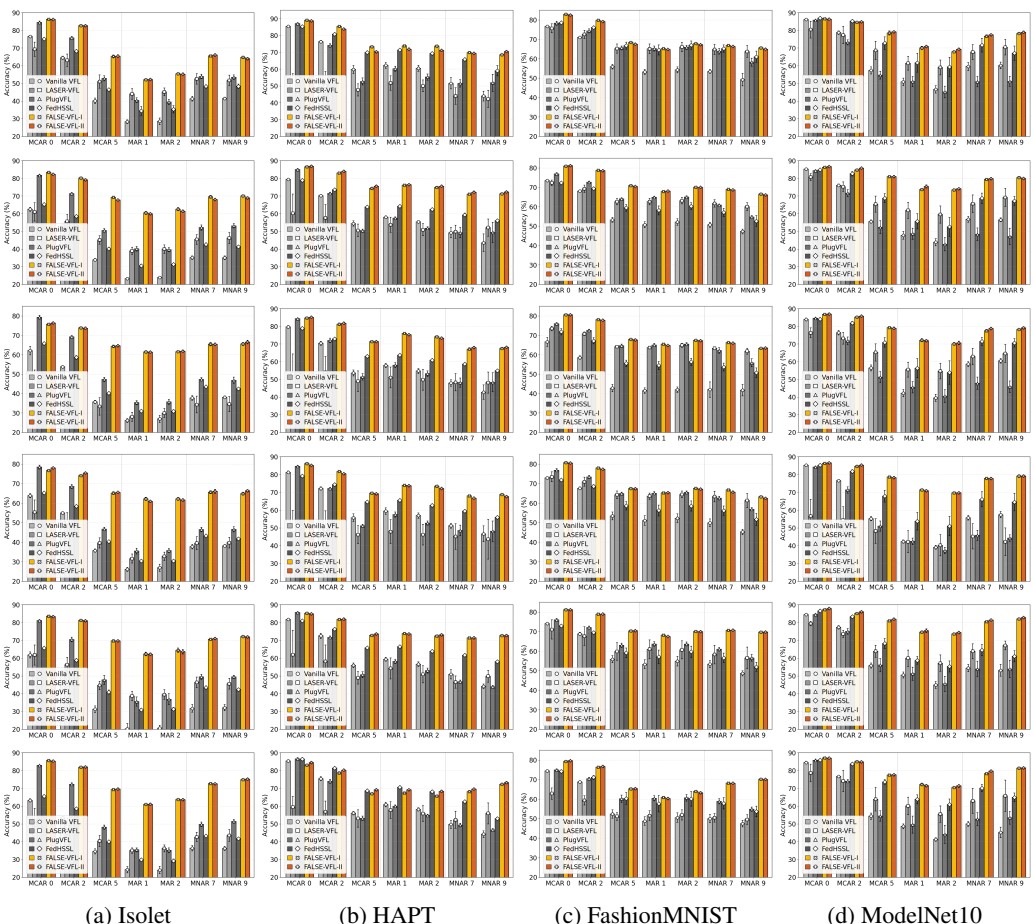

|                | (a) Isolet | (b) HAPT | (c) FashionMNIST | (d) ModelNet10 |
|---|---|---|---|---|

Figure 3: Mean accuracy (%) of six VFL methods trained under six missingness mechanisms and evaluated across seven test patterns. The columns correspond to the datasets: Isolet, HAPT, FashionMNIST, and ModelNet10 from left to right; the rows correspond to the training mechanisms: MCAR 2, MCAR 5, MAR 1, MAR 2, MNAR 7, and MNAR 9 from top to bottom. Bars show mean over five independent runs; error bars show $\pm 1$ standard deviation.

**Robustness to the number of parties.** We further evaluate robustness with respect to the number of parties $K$, varying it from 4 to 12. We do not include MAR in the ablation since the missingness of party $i$ depends on the other parties, so changing $K$ alters the mechanism itself and prevents a fair comparison. We therefore report only MCAR 2 and MNAR 7 in Fig. 4. Recall that we fix a small subset of labeled samples to be fully aligned (e.g., 100 when the total is 500), and the remaining labeled samples follow the specified missingness mechanism. As a result, the expected number of additional fully aligned labeled samples falls quickly as $K$ grows.

Under MCAR 2, Vanilla VFL and FedHSSL tend to decline as $K$ increases because the probability that a labeled sample is simultaneously observed by all parties drops rapidly with larger $K$, and these methods train only on the fully aligned labeled data. Under MNAR 7, the missing rate is high, so beyond the fixed fully aligned labeled subset the expected number of additional fully aligned labeled samples is negligible for $K > 4$. In this regime, these two methods often stay flat or improve as $K$ grows. This effect arises since splitting the same total features across more parties makes a complete loss of informative features much less likely, and the fraction of observed parties per sample varies less around its mean, which makes the fused representation more consistent.

PlugVFL and our method can learn from unaligned labeled samples, so they do not rely only on the fully aligned labeled subset. As $K$ increases they benefit from the same effect, namely a much lower chance that all informative features are missing at once and a more stable observed party fraction per sample, so their performance generally improves or remains stable under both MCAR 2 and MNAR 7.

LASER-VFL can effectively leverage unaligned samples, but assumes a batch-wise missingness mask. As $K$ grows, the space of possible missingness patterns explodes, making it increasingly difficult to assemble mini-batches that share a single mask and effectively lowering batch utilization. Consequently, its performance tends to degrade. Across all configurations, FALSE-VFL consistently outperforms all baselines regardless of the number of parties.

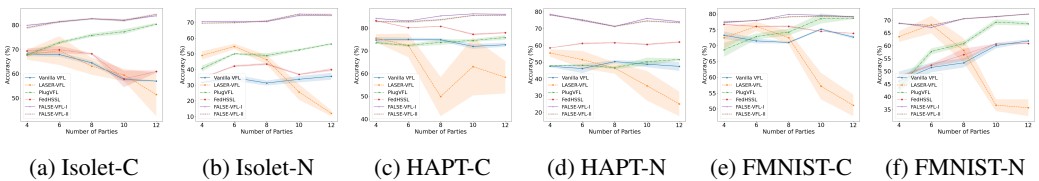

| (a) Isolet-C | (b) Isolet-N | (c) HAPT-C | (d) HAPT-N | (e) FMNIST-C | (f) FMNIST-N |

Figure 4: Mean accuracy (%) with varying numbers of parties for six VFL methods. Panels with suffix "C" (Isolet-C, HAPT-C, FMNIST-C) are trained and evaluated under MCAR 2, and panels with suffix "N" (Isolet-N, HAPT-N, FMNIST-N) are trained and evaluated under MNAR 7. Solid lines show mean accuracy over five independent runs; shaded bands show $\pm 1$ standard deviation.

**Robustness to data heterogeneity.** We provide an additional ablation on data heterogeneity. In an eight-party setting, we draw a party specific missing rate vector $\omega$ from Dirichlet distribution $Dir(\alpha)$ and set the per party missing probabilities to $p = 1.6\,\omega \in [0,1]^8$ so that the average missing rate is 0.2; if any entry of $p$ exceeds 1, we resample $\omega$. we consider $\alpha \in \{\infty, 10, 1, 0.1\}$. When $\alpha = \infty$ we obtain $\omega = (1/8, \cdots, 1/8)$ and hence $p = (0.2, \cdots, 0.2)$, which corresponds to MCAR 2. To quantify heterogeneity, we report the entropy of the sampled $\omega$ for each $\alpha$ in Table 3. Test accuracies are shown in Fig. 5. Across all datasets, LASER-VFL improves as $\alpha$ decreases (i.e., heterogeneity increases). This is expected since uneven per-party missing rates make the mask distribution more concentrated over a smaller set of configurations, which simplifies forming mini-batches that share a single mask and thus improves batch utilization in LASER-VFL. By contrast, Vanilla VFL and FedHSSL decreases as $\alpha$ decreases, since lower entropy reduces the fraction of aligned samples and shrinks the effective training set. FALSE-VFL remains robust, staying flat and highest across all $\alpha$.

Table 3: Entropy of $\omega$ sampled from $Dir(\alpha)$ for each $\alpha$. Larger entropy implies closer to uniform.

| $\alpha$ | $\infty$ | 10 | 1 | 0.1 |
|---|---|---|---|---|
| Isolet | 2.08 | 2.02 | 1.74 | 1.11 |
| HAPT | 2.08 | 2.05 | 1.78 | 1.16 |
| FMNIST | 2.08 | 2.07 | 1.47 | 0.73 |

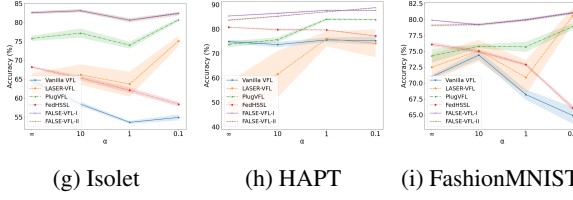

| (g) Isolet | (h) HAPT | (i) FashionMNIST |

Figure 5: Mean accuracy (%) of six VFL methods trained and evaluated under four heterogeneous missingness mechanisms sampled from a Dirichlet distribution. Solid lines show mean accuracy over five independent runs; shaded bands show $\pm 1$ standard deviation.

## 6 CONCLUSIONS

We introduce FALSE-VFL, a vertical federated learning framework that utilizes both unlabeled and unaligned data, supports inference on unaligned data, and accommodates all three missing data mechanisms (MCAR, MAR, MNAR) in both theory and practice. Extensive experiments covering six training and seven test missingness settings show that FALSE-VFL surpasses the existing methods in almost every configuration with a clear gap. Additional ablations on missing rates, the number of parties, and data heterogeneity further validates its robustness.

These findings demonstrate that FALSE-VFL is a practical step toward privacy-preserving collaboration in real-world feature-partitioned settings where labels are scarce and perfect alignment is rare.

ACKNOWLEDGMENTS

This work was supported by the National Research Foundation of Korea (NRF) grant funded by the Korea government (MSIT) (No. RS-2019-NR040050).

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

# A    MISSING DATA MECHANISMS

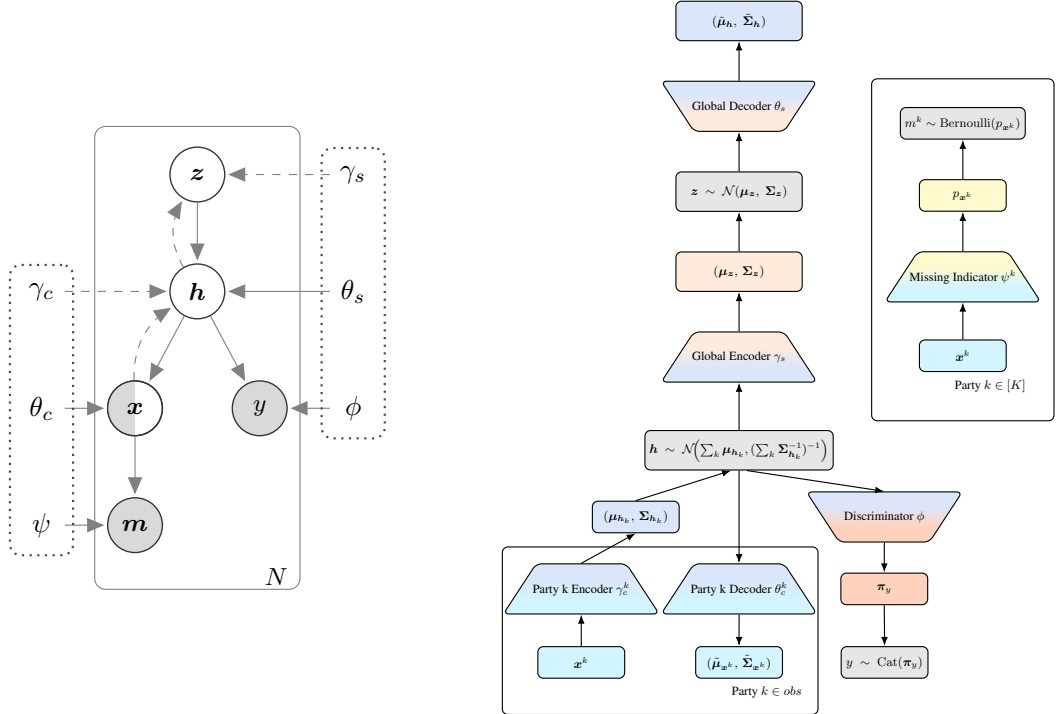

Figure 6: **Left:** Graphical model for FALSE-VFL-II. The left dotted box groups feature-side modules, while the right dotted box groups label-side modules. **Right:** Computational structure for FALSE-VFL-II.

## A.1    MAR MECHANISM

Assume that missing data occurs with the MAR mechanism. Then,

$$
\begin{aligned}
\log p_{\Theta,\psi}(y|\boldsymbol{x}^{obs},\boldsymbol{m}) &= \log p_{\Theta,\psi}(y,\boldsymbol{x}^{obs},\boldsymbol{m}) - \log p_{\psi}(\boldsymbol{x}^{obs},\boldsymbol{m}) \\
&= \log \int p_{\phi}(y|\boldsymbol{h})p_{\psi}(\boldsymbol{m}|\boldsymbol{x}^{obs},\boldsymbol{x}^{mis})p_{\Theta_g}(\boldsymbol{x}^{obs},\boldsymbol{x}^{mis},\boldsymbol{h})\,d\boldsymbol{h}d\boldsymbol{x}^{mis} - \log p_{\psi}(\boldsymbol{x}^{obs},\boldsymbol{m}) \\
&= \log \left( p_{\psi}(\boldsymbol{m}|\boldsymbol{x}^{obs})\int p_{\phi}(y|\boldsymbol{h})p_{\Theta_g}(\boldsymbol{x}^{obs},\boldsymbol{x}^{mis},\boldsymbol{h})\,d\boldsymbol{h}d\boldsymbol{x}^{mis} \right) - \log p_{\psi}(\boldsymbol{x}^{obs},\boldsymbol{m}) \\
&= \log p_{\psi}(\boldsymbol{m}|\boldsymbol{x}^{obs}) + \log p_{\Theta}(y,\boldsymbol{x}^{obs}) - \log p_{\psi}(\boldsymbol{x}^{obs},\boldsymbol{m}) \\
&= \log p_{\Theta}(y|\boldsymbol{x}^{obs}),
\end{aligned}
$$

where the MAR assumption is used in the third equality.

## A.2    MNAR MECHANISM

Under the MNAR mechanism, we need to model the mask explicitly through an additional component $p_{\psi}(\boldsymbol{m}|\boldsymbol{x})$. In addition to the existing models in the MAR case, each party $k \in [K]$ has its own missing indicator parameterized by $\psi^k$ which maps $\boldsymbol{x}^k$ to $p_{\boldsymbol{x}^k}$, a parameter of Bernoulli distribution. Note that $\psi = \{\psi^k\}_{k\in[K]}$ and $p_{\psi}(\boldsymbol{m}|\boldsymbol{x}) = \prod_{k\in[K]} p_{\psi^k}(m^k|\boldsymbol{x}^k)$. The complete computational structure is shown in Fig. 6. Under the graphical model depicted in Fig. 6, we can adopt a procedure

analogous to our original algorithm by first maximizing $\mathcal{L}_\kappa(\Theta_g, \psi)$ as a pretraining step where

$$\sum_{i \in [N]} \log p_{\Theta_g, \psi}(\boldsymbol{x}_i^{obs}, \boldsymbol{m}_i) \geq \mathcal{L}_\kappa(\Theta_g, \psi)$$

$$:= \sum_{i \in [N]} \mathbb{E}_{\{(\boldsymbol{h}_j, \boldsymbol{z}_j, \boldsymbol{x}_j^{mis})\}_{j=1}^\kappa \sim q_{\gamma_c}(\boldsymbol{h}|\boldsymbol{x}_i^{obs}) q_{\gamma_s}(\boldsymbol{z}|\boldsymbol{h}) p_{\theta_c}(\boldsymbol{x}^{mis}|\boldsymbol{h})} \left[ \log R_\kappa(\boldsymbol{x}_i^{obs}, \boldsymbol{m}_i) \right],$$

with

$$R_\kappa(\boldsymbol{x}^{obs}, \boldsymbol{m}) = \frac{1}{\kappa} \sum_{j=1}^\kappa \frac{p_\psi(\boldsymbol{m}|\boldsymbol{x}^{obs}, \boldsymbol{x}_j^{mis}) p_{\theta_c}(\boldsymbol{x}^{obs}|\boldsymbol{h}_j) p_{\theta_s}(\boldsymbol{h}_j|\boldsymbol{z}_j) p(\boldsymbol{z}_j)}{q_{\gamma_c}(\boldsymbol{h}_j|\boldsymbol{x}^{obs}) q_{\gamma_s}(\boldsymbol{z}_j|\boldsymbol{h}_j)}.$$

This follows from

$$\log p_{\Theta_g, \psi}(\boldsymbol{x}^{obs}, \boldsymbol{m}) = \log \int p_\psi(\boldsymbol{m}|\boldsymbol{x}^{obs}, \boldsymbol{x}^{mis}) p_{\theta_c}(\boldsymbol{x}^{obs}|\boldsymbol{h}) p_{\theta_c}(\boldsymbol{x}^{mis}|\boldsymbol{h}) p_{\theta_s}(\boldsymbol{h}|\boldsymbol{z}) p(\boldsymbol{z}) \, d\boldsymbol{h} d\boldsymbol{z} d\boldsymbol{x}^{mis}.$$

We assumed here $p_{\theta_c}(\boldsymbol{x}|\boldsymbol{h})$ is fully factorized, so that $p_{\theta_c}(\boldsymbol{x}|\boldsymbol{h}) = p_{\theta_c}(\boldsymbol{x}^{obs}|\boldsymbol{h}) p_{\theta_c}(\boldsymbol{x}^{mis}|\boldsymbol{h})$.

After the pretraining step, we fix the parameters $\Theta_g, \psi$ and proceed with an analogous training as in our main algorithm. In summary, to handle the MNAR case, we only need an additional model $p_\psi(\boldsymbol{m}|\boldsymbol{x})$ and the sampling of $\boldsymbol{x}^{mis}$.

We also describe what information is communicated between the parties in FALSE-VFL-II.

In the pretraining and training steps, there is an additional exchange for samples with missing parties, in addition to the communication in FALSE-VFL-I. The active party sends sampled latent variables from the global posterior to the missing parties. Each such party uses the received samples with its local decoder and missing indicator to compute the missingness probabilities, and sends them back to the active party.

In the inference step, we follow the same forward communication pattern as in the pretraining and training phases, but no gradients are exchanged.

# B    FEATURE-SIDE CONTRIBUTIONS TO POSTERIOR APPROXIMATION

As defined in Section 3.3, the variational posterior $q_{\gamma_c}(\boldsymbol{h}|\boldsymbol{x}^{obs})$ in (1) is modeled as a Gaussian distribution where both mean and variance are derived from feature-side encoder outputs. This section explains the rationale behind this formulation.

## B.1    PARTY MEAN AGGREGATION

The latent variable $\boldsymbol{h}$ is inferred by aggregating contributions from multiple parties, each providing a mean $\mu_{\gamma_c^k}(\boldsymbol{x}^k)$ based on its local observation $\boldsymbol{x}^k$. The overall posterior mean is computed as the average of these feature-side means, forming a global latent representation. By averaging only the contributions from observed parties, the scale remains consistent regardless of the number of missing parties. Consequently, only the participating parties influence the latent variable.

## B.2    PRECISION-BASED VARIANCE AGGREGATION

The posterior variance is determined by the inverse of the sum of precision matrices $\Sigma_{\gamma_c^k}^{-1}(\boldsymbol{x}^k)$ from each party. As more parties contribute data, the total precision increases, reducing the uncertainty about the latent variable. Parties that do not provide data are implicitly treated as having infinite variance, meaning their absence does not affect the overall precision. This aggregation ensures that as more parties participate, the model becomes more confident (i.e., the variance decreases), leading to a more precise estimate of $\boldsymbol{h}$.

## C  THEORETICAL PROPERTIES

**Theorem C.1.** *Let*

$$\mathcal{L}_\kappa = \mathbb{E}_{\{(\boldsymbol{h}_j, \boldsymbol{z}_j)\}_{j=1}^\kappa \sim q(\boldsymbol{h}|\boldsymbol{x}^{obs})q(\boldsymbol{z}|\boldsymbol{h})} \left[ \log R_\kappa(\boldsymbol{x}^{obs}) \right],$$

$$\mathcal{L}_\kappa^{'} = \mathbb{E}_{\{(\boldsymbol{h}_j, \boldsymbol{z}_j)\}_{j=1}^\kappa \sim q(\boldsymbol{h}|\boldsymbol{x}^{obs})q(\boldsymbol{z}|\boldsymbol{h})} \left[ \log R_\kappa^{'}(y, \boldsymbol{x}^{obs}) \right]$$

*where*

$$R_\kappa(\boldsymbol{x}^{obs}) = \frac{1}{\kappa} \sum_{j=1}^\kappa \frac{p(\boldsymbol{x}^{obs}|\boldsymbol{h}_j)p(\boldsymbol{h}_j|\boldsymbol{z}_j)p(\boldsymbol{z}_j)}{q(\boldsymbol{h}_j|\boldsymbol{x}^{obs})q(\boldsymbol{z}_j|\boldsymbol{h}_j)},$$

$$R_\kappa^{'}(y, \boldsymbol{x}^{obs}) = \frac{1}{\kappa} \sum_{j=1}^\kappa \frac{p(y|\boldsymbol{h}_j)p(\boldsymbol{x}^{obs}|\boldsymbol{h}_j)p(\boldsymbol{h}_j|\boldsymbol{z}_j)p(\boldsymbol{z}_j)}{q(\boldsymbol{h}_j|\boldsymbol{x}^{obs})q(\boldsymbol{z}_j|\boldsymbol{h}_j)}.$$

*Then,* $\mathcal{L}_\kappa$ $\left( \mathcal{L}_\kappa^{'}, \text{resp.} \right)$ *increases as* $\kappa$ *increases, and bounded above by* $\log p(\boldsymbol{x}^{obs})$ $\left( \log p(y, \boldsymbol{x}^{obs}), \text{resp.} \right)$. *In addition, if* $\log \frac{p(\boldsymbol{x}^{obs}, \boldsymbol{h}, \boldsymbol{z})}{q(\boldsymbol{h}, \boldsymbol{z}|\boldsymbol{x}^{obs})}$ $\left( \log \frac{p(y, \boldsymbol{x}^{obs}, \boldsymbol{h}, \boldsymbol{z})}{q(\boldsymbol{h}, \boldsymbol{z}|\boldsymbol{x}^{obs})}, \text{resp.} \right)$ *is bounded, then* $\mathcal{L}_\kappa$ $\left( \mathcal{L}_\kappa^{'}, \text{resp.} \right)$ *converges to* $\log p(\boldsymbol{x}^{obs})$ $\left( \log p(y, \boldsymbol{x}^{obs}), \text{resp.} \right)$ *as* $k \to \infty$.

*Proof.* We apply Theorem 1 in Burda et al. (2016). Here, we give the proof for $\mathcal{L}_\kappa^{'}$. We can prove the upper bound using Jensen's inequality as

$$\mathcal{L}_\kappa^{'} = \mathbb{E}_{\{(\boldsymbol{h}_j, \boldsymbol{z}_j)\}_{j=1}^\kappa \sim q(\boldsymbol{h}|\boldsymbol{x}^{obs})q(\boldsymbol{z}|\boldsymbol{h})} \left[ \log \frac{1}{\kappa} \sum_{j=1}^\kappa \frac{p(y|\boldsymbol{h}_j)p(\boldsymbol{x}^{obs}|\boldsymbol{h}_j)p(\boldsymbol{h}_j|\boldsymbol{z}_j)p(\boldsymbol{z}_j)}{q(\boldsymbol{h}_j|\boldsymbol{x}^{obs})q(\boldsymbol{z}_j|\boldsymbol{h}_j)} \right]$$

$$\leq \log \mathbb{E}_{\{(\boldsymbol{h}_j, \boldsymbol{z}_j)\}_{j=1}^\kappa \sim q(\boldsymbol{h}|\boldsymbol{x}^{obs})q(\boldsymbol{z}|\boldsymbol{h})} \left[ \frac{1}{\kappa} \sum_{j=1}^\kappa \frac{p(y, \boldsymbol{x}^{obs}, \boldsymbol{h}_j, \boldsymbol{z}_j)}{q(\boldsymbol{h}_j|\boldsymbol{x}^{obs})q(\boldsymbol{z}_j|\boldsymbol{h}_j)} \right] = \log p(y, \boldsymbol{x}^{obs}).$$

To prove monotonic increase, let $I$ be a uniformly chosen subset of size $m$ from $[\kappa]$. Using Jensen's inequality again, we get

$$\mathcal{L}_\kappa^{'} = \mathbb{E}_{\{(\boldsymbol{h}_j, \boldsymbol{z}_j)\}_{j=1}^\kappa} \left[ \log \frac{1}{\kappa} \sum_{j=1}^\kappa \frac{p(y, \boldsymbol{x}^{obs}, \boldsymbol{h}_j, \boldsymbol{z}_j)}{q(\boldsymbol{h}_j, \boldsymbol{z}_j|\boldsymbol{x}^{obs})} \right]$$

$$= \mathbb{E}_{\{(\boldsymbol{h}_j, \boldsymbol{z}_j)\}_{j=1}^\kappa} \left[ \log \mathbb{E}_I \left[ \frac{1}{m} \sum_{j=1}^m \frac{p(y, \boldsymbol{x}^{obs}, \boldsymbol{h}_j, \boldsymbol{z}_j)}{q(\boldsymbol{h}_j, \boldsymbol{z}_j|\boldsymbol{x}^{obs})} \right] \right]$$

$$\geq \mathbb{E}_{\{(\boldsymbol{h}_j, \boldsymbol{z}_j)\}_{j=1}^\kappa} \left[ \mathbb{E}_I \left[ \log \frac{1}{m} \sum_{j=1}^m \frac{p(y, \boldsymbol{x}^{obs}, \boldsymbol{h}_j, \boldsymbol{z}_j)}{q(\boldsymbol{h}_j, \boldsymbol{z}_j|\boldsymbol{x}^{obs})} \right] \right]$$

$$= \mathbb{E}_{\{(\boldsymbol{h}_j, \boldsymbol{z}_j)\}_{j=1}^m} \left[ \log \frac{1}{m} \sum_{j=1}^m \frac{p(y, \boldsymbol{x}^{obs}, \boldsymbol{h}_j, \boldsymbol{z}_j)}{q(\boldsymbol{h}_j, \boldsymbol{z}_j|\boldsymbol{x}^{obs})} \right] = \mathcal{L}_m^{'}.$$

Lastly, assume that $\log \frac{p(y, \boldsymbol{x}^{obs}, \boldsymbol{h}, \boldsymbol{z})}{q(\boldsymbol{h}, \boldsymbol{z}|\boldsymbol{x}^{obs})}$ is bounded. Then, $R_\kappa^{'}(y, \boldsymbol{x}^{obs})$ converges to $p(y, \boldsymbol{x}^{obs})$ by the strong law of large numbers. Take log on both sides and by the dominated convergence theorem, $\mathcal{L}_\kappa^{'}$ converges to $\log p(y, \boldsymbol{x}^{obs})$. □

## D  EXPERIMENTAL DETAILS

### D.1  BASELINES

To ensure a fair comparison, we apply a unified two-stage protocol to the baselines that do not natively exploit unlabeled data: Vanilla VFL, LASER-VFL, and PlugVFL. First, every local model

is pretrained with SimSiam (Chen & He, 2021), one of the representative self-supervised learning methods, which He et al. (2024) report to outperform BYOL and MoCo on VFL benchmarks. The pretrained networks are then finetuned on the available labeled subset. The baselines considered are summarized below.

**Vanilla VFL:** Since Vanilla VFL requires fully aligned data, we train it only on aligned data. For prediction, we employ the simple zero-imputation strategy for unaligned data which is more effective than a random prediction.

**LASER-VFL (Valdeira et al., 2024):** In the original LASER-VFL, each party has its own representation model and a fusion model. However, the fusion model only works for parties that hold labels. Since we consider the single active party scenario, which is the most general setting in VFL framework, we employ a version of LASER-VFL with only one fusion model.

**PlugVFL (Sun et al., 2024):** PlugVFL was originally designed for fully aligned data in training, so we again use zero-imputation for unaligned parties in both training and inference. We set $p = 0.5$ for the probability of dropping each passive party, and disable the label IP protection objective which enhances label privacy but can reduce performance. Since this version of PlugVFL is well implemented in the work of LASER-VFL, we adopt it.

**FedHSSL (He et al., 2024):** FedHSSL supports three SSL methods: SimSiam, BYOL, and MoCo. Since SimSiam achieves the best performance in the authors' experiments, we adopt it in all runs. FedHSSL also need fully aligned data for finetuning, so we apply zero-imputation for inference similarly.

### D.2 DATASETS AND MODELS

**Datasets.** **Isolet** is a speech recognition dataset containing 7,797 audio recordings of 150 speakers pronouncing each 26-letter English alphabet twice. Each recording is represented by 617 acoustic features extracted from the raw audio waveform. We partition these 617 features evenly across eight parties by assigning each party 77 features (one feature is discarded).

**HAPT** (Human Activities and Postural Transitions) dataset comprises smartphone accelerometer and gyroscope signals collected from 30 volunteers performing 12 daily activities. Each data sample is represented as a 561-dimensional feature vector. We evenly distribute these features across eight parties, assigning each party 70 features (one feature is discarded).

**FashionMNIST** is a widely-used benchmark consisting of grayscale images of fashion items divided into 10 classes. For VFL settings, we partition each $28 \times 28$ image into eight segments of size $14 \times 7$, with each segment assigned to one of the eight parties.

**ModelNet10** is a dataset of 3,991 training and 908 test samples of 3D CAD models belonging to 10 classes, commonly used in multi-view shape recognition. In our experiments, each 3D model is converted into 12 distinct 2D views by rotating the object $360°$, capturing one view every $30°$. To create a challenging VFL task, we group adjacent pairs of views, forming six pairs that correspond to six parties. For each sample, we randomly select one view from each pair, providing each party with a single $224 \times 224$ image per sample. To further increase task difficulty, images are resized to $32 \times 32$ pixels. This random selection process is repeated 6 times for the training set and 2 times for the test set, resulting in 23,946 training samples and 1,816 test samples.

**Models.** Tables 4 to 6 show detailed model architectures for each baseline. Tables 7 to 11 report hyperparameters used in our experiments.

Table 4: Model architecture details for each component of Vanilla VFL, LASER-VFL, and PlugVFL.

| Component | Structure |
|---|---|
| Feature Extractor | ResNet-18 or 3-layer MLP |
| Projector | 3-layer MLP |
| Predictor | 2-layer MLP |
| Discriminator | 2-layer MLP |

Table 5: Model architecture details for each component of FedHSSL.

| Component | Structure |
|---|---|
| Local Bottom Encoder | Lower layers of ResNet-18 or 1-layer MLP |
| Local Top Encoder | Upper layers of ResNet-18 or 1-layer MLP |
| Cross-Party Encoder | ResNet-18 or 2-layer MLP |
| Projector | 3-layer MLP |
| Predictor | 2-layer MLP |
| Discriminator | 2-layer MLP |

Table 6: Model architecture details for each component of FALSE-VFL.

| Component | Structure |
|---|---|
| Party Encoder ($\gamma_c^k$) | ResNet-18 or 2-layer MLP |
| Party Decoder ($\theta_c^k$) | Transposed CNN or 2-layer MLP |
| Party Missing Indicator ($\psi^k$) | 3-layer MLP |
| Global Encoder ($\gamma_s$) | 3-layer MLP |
| Global Decoder ($\theta_s$) | 3-layer MLP |
| Discriminator ($\phi$) | 2-layer MLP |

Table 7: Hyperparameters for Vanilla VFL. Values for pretraining are shown in parentheses.

| Hyperparameter | Isolet | HAPT | FashionMNIST | ModelNet10 |
|---|---|---|---|---|
| Optimizer | Adam (SGD) | Adam (SGD) | Adam (SGD) | Adam (SGD) |
| Learning Rate | 5e-4 (0.025) | 5e-3 (0.02) | 1e-4 (0.02) | 1e-4 (0.025) |
| Batch Size | 32 (512) | 64 (512) | 128 (1024) | 128 (1024) |
| Epochs | 500 (100) | 100 (150) | 150 (100) | 100 (50) |
| Weight Decay | 1e-4 (3e-5) | 1e-4 (3e-5) | 1e-4 (3e-5) | 1e-4 (3e-5) |
| Latent Dimension | 128 | 128 | 196 | 256 |

Table 8: Hyperparameters for LASER-VFL. Values for pretraining are shown in parentheses.

| Hyperparameter | Isolet | HAPT | FashionMNIST | ModelNet10 |
|---|---|---|---|---|
| Optimizer | Adam (SGD) | Adam (SGD) | Adam (SGD) | Adam (SGD) |
| Learning Rate | 5e-5 (0.025) | 1e-3 (0.02) | 1e-4 (0.02) | 5e-4 (0.025) |
| Batch Size | 64 (512) | 128 (512) | 128 (1024) | 128 (1024) |
| Epochs | 200 (100) | 100 (150) | 100 (100) | 10 (50) |
| Weight Decay | 1e-4 (3e-5) | 1e-4 (3e-5) | 1e-4 (3e-5) | 1e-4 (3e-5) |
| Latent Dimension | 128 | 128 | 196 | 256 |

Table 9: Hyperparameters for PlugVFL. Values for pretraining are shown in parentheses.

| Hyperparameter | Isolet | HAPT | FashionMNIST | ModelNet10 |
|---|---|---|---|---|
| Optimizer | Adam (SGD) | Adam (SGD) | Adam (SGD) | Adam (SGD) |
| Learning Rate | 5e-5 (0.025) | 5e-4 (0.02) | 1e-4 (0.02) | 1e-4 (0.025) |
| Batch Size | 32 (512) | 64 (512) | 128 (1024) | 128 (1024) |
| Epochs | 100 (100) | 100 (150) | 150 (100) | 50 (50) |
| Weight Decay | 1e-4 (3e-5) | 1e-4 (3e-5) | 1e-4 (3e-5) | 1e-4 (3e-5) |
| Latent Dimension | 128 | 128 | 196 | 256 |

Table 10: Hyperparameters for FedHSSL. Values for pretraining are shown in parentheses.

| Hyperparameter | Isolet | HAPT | FashionMNIST | ModelNet10 |
|---|---|---|---|---|
| Optimizer | Adam (SGD) | Adam (Adam) | Adam (SGD) | Adam (SGD) |
| Learning Rate | 2e-3 (0.025) | 5e-4 (0.025) | 1e-4 (0.02) | 5e-4 (0.025) |
| Batch Size | 64 (512) | 64 (512) | 128 (1024) | 128 (1024) |
| Epochs | 300 (150) | 500 (100) | 100 (150) | 100 (150) |
| Weight Decay | 1e-4 (3e-5) | 1e-4 (1e-5) | 1e-4 (3e-5) | 1e-4 (3e-5) |
| Latent Dimension | 128 | 128 | 196 | 256 |

Table 11: Hyperparameters for FALSE-VFL. Values for pretraining are shown in parentheses. We use $\kappa = 10$ and $L = 50$ for all datasets.

| Hyperparameter | Isolet | HAPT | FashionMNIST | ModelNet10 |
|---|---|---|---|---|
| Optimizer | Adam (Adam) | Adam (Adam) | Adam (Adam) | Adam (Adam) |
| Learning Rate | 2e-4 (5e-4) | 2e-4 (2e-3) | 2e-4 (5e-5) | 2e-4 (1e-4) |
| Batch Size | 128 (512) | 128 (512) | 128 (1024) | 128 (1024) |
| Epochs | 300 (300) | 300 (500) | 200 (150) | 200 (300) |
| Weight Decay | 1e-4 (1e-4) | 1e-4 (1e-4) | 1e-4 (1e-4) | 1e-4 (1e-4) |
| $h$ Dimension | 128 | 128 | 196 | 256 |
| $z$ Dimension | 64 | 64 | 32 | 64 |

## D.3 ALGORITHM

---

**Algorithm 1** FALSE-VFL-I: two-stage training and inference

---

1: **Hyperparameters:** $K$: the number of parties, $\kappa/L$: the number of importance samples for training / test, $\eta_{\text{pre}}/\eta_{\text{train}}$: learning rates for pretraining / training, $T_{\text{pre}}/T_{\text{train}}$: epochs for pretraining / training

2: **Parameters:** $\gamma_c^k/\theta_c^k$: encoder / decoder for party $k \in [K]$, $\gamma_s/\theta_s/\phi$: encoder / decoder / discriminator for active party, $\gamma_c = \{\gamma_c^1, \cdots, \gamma_c^K\}, \theta_c = \{\theta_c^1, \cdots, \theta_c^K\}, \Theta_g = \{\gamma_c, \theta_c, \gamma_s, \theta_s\}, \Theta = \{\Theta_g, \phi\}$

3:

4: *Stage 1 – Pretraining: maximize marginal likelihood* $p_{\Theta_g}(\boldsymbol{x}^{obs})$

5: **for** $t = 1$ **to** $T_{\text{pre}}$ **do**

6:     **for all** minibatch $\mathcal{B}$ **do**

7:         **for all** party $k \in \{\text{obs}\}$ **in parallel do**

8:             $(\boldsymbol{\mu}_k, \boldsymbol{\Sigma}_k) \leftarrow \text{ENC}_{\gamma_c^k}(\boldsymbol{x}_{\mathcal{B}}^k)$

9:             Send $(\boldsymbol{\mu}_k, \boldsymbol{\Sigma}_k)$ to active party

10:         **end for**

11:         Active party forms $q_{\gamma_c}(\boldsymbol{h}|\mathcal{B})$ via Eq. (1); sample $\{(\boldsymbol{h}_j, \boldsymbol{z}_j)\}_{j=1}^{\kappa}$ from $q_{\gamma_c}(\boldsymbol{h}|\mathcal{B})q_{\gamma_s}(\boldsymbol{z}|\boldsymbol{h})$

12:         Compute $\mathcal{L}_\kappa(\Theta_g)$ and update $\Theta_g \leftarrow \Theta_g + \eta_{\text{pre}}\nabla_{\Theta_g}\mathcal{L}_\kappa$

13:     **end for**

14: **end for**

15: Freeze $\Theta_g$

16:

17: *Stage 2 – Training: maximize conditional likelihood* $p_\Theta(y|\boldsymbol{x}^{obs})$

18: **for** $t = 1$ **to** $T_{\text{train}}$ **do**

19:     **for all** labeled minibatch $\mathcal{B}$ **do**

20:         **for all** party $k \in \{\text{obs}\}$ **in parallel do**

21:             $(\boldsymbol{\mu}_k, \boldsymbol{\Sigma}_k) \leftarrow \text{ENC}_{\gamma_c^k}(\boldsymbol{x}_{\mathcal{B}}^k)$

22:             Send $(\boldsymbol{\mu}_k, \boldsymbol{\Sigma}_k)$ to active party

23:         **end for**

24:         Active party forms $q_{\gamma_c}(\boldsymbol{h}|\mathcal{B})$ via Eq. (1); sample $\{(\boldsymbol{h}_j, \boldsymbol{z}_j)\}_{j=1}^{\kappa}$ from $q_{\gamma_c}(\boldsymbol{h}|\mathcal{B})q_{\gamma_s}(\boldsymbol{z}|\boldsymbol{h})$

25:         Compute $\mathcal{L}'_\kappa(\phi)$ and update $\phi \leftarrow \phi + \eta_{\text{train}}\nabla_\phi\mathcal{L}'_\kappa$

26:     **end for**

27: **end for**

28:

29: *Inference on a new incomplete sample* $\boldsymbol{x}^{obs}$

30: Gather $(\boldsymbol{\mu}_k, \boldsymbol{\Sigma}_k)$ from observed parties; sample $\{(\boldsymbol{h}_\ell, \boldsymbol{z}_\ell)\}_{\ell=1}^L$ from $q$

31: Compute importance weights $w_\ell$ and predict $\hat{y} = \sum_{\ell=1}^L w_\ell \, p_\phi(y|\boldsymbol{h}_\ell)$

---

## D.4 MAR MECHANISMS

We introduce two MAR mechanisms inspired by real-world scenarios in which the decision to collect further observations depend on the information of previously observed data.

For instance, a patient may choose whether to visit additional hospitals after reviewing one hospital's examination results. Likewise, an individual viewing an image piece by piece might stop as soon as the observed portion is sufficiently informative. In our simulation, we measure "information" by the variance of a piece: low variance implies a near-uniform region (hence little information), whereas high variance implies richer details. All datasets are normalized to ensure scale consistency across features.

**Type 1: Stop at the First Highly Informative Piece.** We begin by randomly selecting one piece. If its variance exceeds a predefined threshold, we consider it "sufficiently informative" and do not observe any additional pieces. Otherwise, we slightly lower the threshold and randomly select another piece. See Algorithm 2 for details.

**Type 2: Accumulate Multiple Moderately Informative Pieces.** We start with a variance threshold $T$ and an "excessive variance" budget $B$. Whenever the variance $v$ of an observed piece exceeds $T$, we subtract $(v - T)$ from $B$. If $B$ falls below zero, we stop observing further pieces. Otherwise, we reduce $T$ slightly and continue. Conceptually, this simulates gathering several moderately informative pieces until reaching a certain limit. See Algorithm 3 for details.

These procedures systematically generate partially aligned data across multiple parties by emulating natural decision processes. To the best of our knowledge, no prior work has addressed MAR-based alignment in the VFL setting. Our methods are designed to reflect realistic user behaviors and can be viewed as a concrete instantiation of the "Threshold method" described in Zhou et al. (2024).

---

**Algorithm 2** MAR Mechanism Type 1: Single High-Informative Piece

---

**Require:** A set of parties (pieces) $P$; initial variance threshold $T = 1.1$; threshold decrement $\Delta = 0.15$.
 1: Choose an initial piece randomly from $P$.
 2: **while** there are unvisited pieces in $P$ **do**
 3:     Compute variance $v$ of the chosen piece.
 4:     **if** $v > T$ **then**
 5:         **Stop** (no more pieces are observed).
 6:     **else**
 7:         $T \leftarrow T - \Delta$ {Lower the threshold}
 8:         Randomly choose the next unvisited piece from $P$.
 9:     **end if**
10: **end while**

---

**Algorithm 3** MAR Mechanism Type 2: Multiple Moderate-Informative Pieces

---

**Require:** A set of parties (pieces) $P$; initial variance threshold $T = 0.5$; total "excessive variance" budget $B = 0.7(0.5$ for ModelNet10); threshold decrement $\Delta = 0.15$.
 1: Choose an initial piece randomly from $P$.
 2: **while** there are unvisited pieces in $P$ **do**
 3:     Compute variance $v$ of the chosen piece.
 4:     **if** $v > T$ **then**
 5:         $B \leftarrow B - (v - T)$ {Consume part of the budget}
 6:     **end if**
 7:     **if** $B \leq 0$ **then**
 8:         **Stop** (no more pieces are observed).
 9:     **else**
10:         $T \leftarrow T - \Delta$ {Lower the threshold}
11:         Randomly choose the next unvisited piece from $P$.
12:     **end if**
13: **end while**

---

## D.5 IMPLEMENTATION

The experiments are implemented in PyTorch. We simulate a decentralized environment using a single deep learning workstation equipped with an Intel(R) Xeon(R) Gold 6348 CPU, one NVIDIA GeForce RTX 3090 GPU, and 263 GB of RAM. The runtime of FALSE-VFL for each dataset is reported in Table 12.

Table 12: Execution time (in minutes) of FALSE-VFL on each dataset. Times are separated into pretraining and main training phases.

| Dataset | Pretraining Time (min) | Training Time (min) |
|---|---|---|
| Isolet | 3 | 2 |
| HAPT | 6 | 2 |
| FashionMNIST | 60 | 30 |
| ModelNet10 | 860 | 90 |

## E ADDITIONAL EXPERIMENTAL RESULTS

Table 13: Mean accuracy (%) and standard deviation (in parentheses) over five independent runs for six VFL methods trained under MCAR 2 conditions and evaluated on seven test patterns. Boldface highlights the best result in each column. An asterisk ($^*$) marks accuracy computed with labeled data only, as including unlabeled data led to lower accuracy.

| Test Data | MCAR 0 | MCAR 2 | MCAR 5 | MAR 1 | MAR 2 | MNAR 7 | MNAR 9 |
|---|---|---|---|---|---|---|---|
| | | | Isolet | | | | |
| Vanilla VFL | 76.6 (0.3) | 64.5 (0.9) | 40.1 (1.6)$^*$ | 28.3 (1.0)$^*$ | 28.5 (1.9)$^*$ | 41.3 (1.4)$^*$ | 41.4 (0.8)$^*$ |
| LASER-VFL | 69.3 (4.0) | 63.1 (3.5) | 51.0 (3.8) | 44.0 (3.0) | 45.1 (2.2) | 52.5 (2.9) | 51.7 (2.9) |
| PlugVFL | 84.5 (0.7) | 75.8 (0.7) | 52.8 (1.4) | 40.4 (1.4) | 39.4 (1.5) | 53.9 (1.2) | 53.5 (1.3) |
| FedHSSL | 75.1 (0.3) | 68.2 (0.3) | 46.5 (0.6) | 34.4 (2.5)$^*$ | 35.3 (2.1)$^*$ | 48.2 (0.8) | 48.3 (1.2) |
| **FALSE-VFL-I** | **86.3** (0.1) | **82.6** (0.4) | 65.2 (0.9) | 52.0 (0.5) | **55.4** (0.5) | 65.5 (0.5) | **64.7** (0.7) |
| **FALSE-VFL-II** | 86.2 (0.2) | **82.6** (0.3) | **65.4** (0.8) | **52.2** (0.6) | 55.2 (0.7) | **66.0** (0.7) | 64.0 (0.8) |
| | | | HAPT | | | | |
| Vanilla VFL | 85.4 (0.3) | 76.3 (0.8)$^*$ | 59.7 (2.4)$^*$ | 62.2 (1.7)$^*$ | 60.4 (1.8)$^*$ | 51.5 (3.3)$^*$ | 43.9 (2.9)$^*$ |
| LASER-VFL | 47.3 (9.9) | 49.9 (8.6) | 47.6 (3.6) | 51.7 (4.3) | 50.0 (3.5) | 43.9 (4.9)$^*$ | 42.2 (5.1)$^*$ |
| PlugVFL | 87.0 (0.4) | 74.0 (1.0)$^*$ | 52.6 (2.1) | 60.1 (1.5) | 55.2 (1.9) | 51.4 (2.2)$^*$ | 51.8 (5.1)$^*$ |
| FedHSSL | 85.3 (0.2) | 80.8 (0.2) | 69.8 (0.3) | 71.4 (0.4) | 69.3 (0.6) | 65.7 (0.5) | 59.0 (3.1) |
| **FALSE-VFL-I** | **89.0** (0.2) | **85.4** (0.2) | **73.3** (0.4) | **73.8** (0.4) | **73.7** (0.3) | **69.8** (0.6) | 68.5 (0.3) |
| **FALSE-VFL-II** | 88.6 (0.1) | 83.7 (0.4) | 70.3 (0.4) | 71.8 (0.7) | 71.0 (0.3) | 69.3 (0.4) | **70.4** (0.4) |
| | | | FashionMNIST | | | | |
| Vanilla VFL | 76.7 (0.2) | 71.0 (0.3) | 55.9 (1.2) | 53.2 (1.3) | 54.2 (1.5) | 53.4 (1.0) | 49.2 (3.2)$^*$ |
| LASER-VFL | 75.6 (2.3) | 72.5 (2.1) | 65.2 (2.4) | 65.1 (2.5) | 66.0 (2.4) | 64.7 (2.6) | 63.7 (3.0) |
| PlugVFL | 78.3 (1.0) | 74.3 (1.1) | 65.5 (1.0) | **65.2** (1.3) | 65.8 (1.1) | 63.8 (1.4) | 58.2 (2.1) |
| FedHSSL | 78.5 (0.2) | 76.1 (0.3) | 66.7 (1.8) | 64.5 (2.6) | 67.0 (2.1) | 65.2 (1.9) | 61.0 (2.9) |
| **FALSE-VFL-I** | **82.9** (0.1) | **79.9** (0.1) | **68.4** (0.3) | **65.2** (0.3) | **67.8** (0.4) | **66.8** (0.3) | **65.6** (0.4) |
| **FALSE-VFL-II** | 82.5 (0.3) | 79.1 (0.2) | 67.5 (0.3) | 64.7 (0.3) | 67.2 (0.3) | 66.3 (0.3) | 64.9 (0.3) |
| | | | ModelNet10 | | | | |
| Vanilla VFL | 86.1 (0.6) | 78.7 (1.0) | 57.3 (2.0) | 50.8 (2.1) | 46.6 (2.1) | 59.5 (2.3) | 60.1 (1.9) |
| LASER-VFL | 80.4 (4.7) | 77.3 (4.9) | 68.6 (5.2) | 61.4 (3.9) | 59.0 (4.2) | 67.7 (4.2) | 70.5 (4.4) |
| PlugVFL | 85.7 (0.5) | 73.2 (1.4) | 54.6 (2.2) | 51.0 (2.9) | 45.1 (3.3) | 50.9 (2.9) | 51.1 (2.9) |
| FedHSSL | **86.9** (0.4) | **85.2** (0.5) | 72.8 (3.5) | 61.6 (5.3) | 59.3 (5.4) | 71.6 (3.9) | 67.1 (4.0) |
| **FALSE-VFL-I** | 86.6 (0.2) | 84.6 (0.2) | 78.5 (1.2) | 70.1 (0.9) | 67.8 (0.4) | 76.7 (0.7) | 78.3 (0.6) |
| **FALSE-VFL-II** | 86.3 (0.3) | 84.8 (0.2) | **79.1** (0.4) | **70.8** (0.6) | **69.3** (0.4) | **77.4** (0.5) | **78.9** (0.6) |

Table 14: Mean accuracy (%) and standard deviation (in parentheses) over five independent runs for six VFL methods trained under MCAR 5 conditions and evaluated on seven test patterns. Boldface highlights the best result in each column. An asterisk ($^*$) marks accuracy computed with labeled data only, as including unlabeled data led to lower accuracy.

| Test Data | MCAR 0 | MCAR 2 | MCAR 5 | MAR 1 | MAR 2 | MNAR 7 | MNAR 9 |
|---|---|---|---|---|---|---|---|
| | | | | Isolet | | | |
| Vanilla VFL | 62.5 (1.1) | 53.1 (1.1)$^*$ | 33.9 (0.7)$^*$ | 23.2 (0.6)$^*$ | 23.9 (0.4)$^*$ | 35.1 (1.0)$^*$ | 35.0 (1.0)$^*$ |
| LASER-VFL | 61.0 (5.2) | 55.6 (4.0) | 45.1 (2.4) | 38.9 (2.1) | 39.9 (2.4) | 45.4 (2.7) | 46.2 (3.1) |
| PlugVFL | 81.7 (0.5) | 71.3 (0.6) | 50.5 (1.0) | 40.1 (1.3) | 39.6 (1.2) | 52.1 (1.5) | 53.1 (1.2) |
| FedHSSL | 65.2 (0.5) | 58.5 (0.5) | 40.1 (0.9) | 30.5 (0.9) | 31.2 (0.7) | 42.6 (0.7) | 41.3 (0.7) |
| **FALSE-VFL-I** | **83.4** (0.2) | **80.2** (0.5) | **69.3** (0.5) | **60.4** (0.9) | **62.5** (0.9) | **69.5** (0.8) | **70.0** (0.4) |
| **FALSE-VFL-II** | 82.2 (0.2) | 79.2 (0.5) | 67.7 (0.2) | 59.9 (0.4) | 61.4 (0.6) | 68.0 (0.8) | 68.9 (0.6) |
| | | | | HAPT | | | |
| Vanilla VFL | 79.4 (0.5)$^*$ | 70.2 (0.6)$^*$ | 54.4 (1.5)$^*$ | 58.0 (0.8)$^*$ | 55.4 (0.7)$^*$ | 49.2 (2.9)$^*$ | 43.5 (5.0)$^*$ |
| LASER-VFL | 60.2 (10.9) | 57.6 (7.5) | 50.5 (3.6) | 53.7 (4.3) | 50.9 (3.4) | 49.8 (3.7) | 52.1 (4.9) |
| PlugVFL | 85.0 (0.5) | 71.5 (0.7)$^*$ | 50.3 (0.9)$^*$ | 57.5 (0.6) | 51.8 (0.8) | 49.2 (2.9)$^*$ | 49.4 (5.8)$^*$ |
| FedHSSL | 78.9 (0.2) | 73.5 (0.4) | 63.8 (0.4) | 64.2 (0.4) | 62.4 (0.5) | 59.3 (1.0)$^*$ | 56.0 (0.6)$^*$ |
| **FALSE-VFL-I** | 86.5 (0.3) | 83.0 (0.2) | 74.3 (0.7) | 76.1 (0.4) | 74.9 (0.6) | 71.1 (0.3) | 71.2 (0.2) |
| **FALSE-VFL-II** | **86.8** (0.5) | **84.0** (0.3) | **75.5** (0.5) | **76.4** (0.3) | **75.5** (0.3) | **72.2** (0.4) | **72.3** (0.3) |
| | | | | FashionMNIST | | | |
| Vanilla VFL | 73.5 (0.2) | 68.0 (0.3) | 53.2 (1.2) | 50.7 (1.5) | 52.1 (1.6) | 50.7 (1.3) | 47.2 (1.2) |
| LASER-VFL | 72.3 (1.4)$^*$ | 69.2 (1.9) | 62.6 (1.6) | 62.6 (1.6) | 63.1 (1.7) | 61.7 (1.9) | 60.0 (2.0) |
| PlugVFL | 76.9 (0.6) | 72.7 (0.4) | 64.0 (0.5) | 64.8 (0.5) | 65.0 (0.5) | 60.8 (0.5) | 54.7 (0.7) |
| FedHSSL | 72.3 (0.5) | 69.3 (0.7) | 59.4 (1.7) | 58.1 (2.3) | 60.3 (1.7) | 57.0 (2.1) | 52.6 (2.8) |
| **FALSE-VFL-I** | 80.9 (0.1) | **78.8** (0.2) | **70.9** (0.2) | 67.8 (0.3) | **70.0** (0.2) | **69.0** (0.2) | **66.5** (0.2) |
| **FALSE-VFL-II** | **81.2** (0.2) | 78.6 (0.2) | 70.4 (0.2) | **68.0** (0.3) | 69.9 (0.2) | 68.6 (0.0) | 66.0 (0.1) |
| | | | | ModelNet10 | | | |
| Vanilla VFL | 85.2 (0.3) | 76.0 (0.8) | 55.5 (0.9) | 47.7 (2.2) | 43.8 (2.0) | 56.8 (1.6) | 56.5 (1.1) |
| LASER-VFL | 80.7 (2.0) | 75.3 (2.7) | 65.4 (4.5) | 61.9 (4.5) | 59.5 (4.9) | 65.6 (5.0) | 69.2 (5.1) |
| PlugVFL | 84.3 (0.4) | 71.6 (1.9) | 52.3 (3.7) | 48.6 (3.2) | 42.7 (3.8) | 48.3 (3.6) | 47.1 (4.7) |
| FedHSSL | 84.9 (0.5) | 82.6 (1.2) | 68.8 (2.5) | 55.7 (4.3) | 52.5 (5.3) | 68.7 (2.9) | 67.2 (2.5) |
| **FALSE-VFL-I** | 86.3 (0.3) | 84.7 (0.2) | **80.9** (0.3) | 73.7 (0.5) | 73.4 (0.6) | 79.4 (0.5) | **80.5** (0.6) |
| **FALSE-VFL-II** | **86.7** (0.2) | **85.8** (0.2) | **80.9** (0.6) | **75.4** (0.9) | **74.1** (0.4) | **79.8** (0.6) | 79.9 (0.4) |

Table 15: Mean accuracy (%) and standard deviation (in parentheses) over five independent runs for six VFL methods trained under MAR 1 conditions and evaluated on seven test patterns.

| Test Data | MCAR 0 | MCAR 2 | MCAR 5 | MAR 1 | MAR 2 | MNAR 7 | MNAR 9 |
|---|---|---|---|---|---|---|---|
| | | | | Isolet | | | |
| Vanilla VFL | 62.1 (2.2) | 53.8 (0.5)$^*$ | 35.7 (0.8)$^*$ | 26.1 (1.0)$^*$ | 26.9 (1.8)$^*$ | 37.7 (1.2)$^*$ | 38.1 (0.7)$^*$ |
| LASER-VFL | 40.8 (11.3) | 38.6 (8.3) | 33.3 (4.5) | 28.0 (2.3) | 29.9 (2.3) | 34.3 (4.3) | 34.5 (3.9) |
| PlugVFL | **79.3** (1.0) | 69.2 (0.6) | 47.2 (1.1) | 35.3 (1.0) | 35.7 (1.3) | 47.3 (0.9) | 46.8 (1.6) |
| FedHSSL | 65.8 (0.6) | 58.6 (0.9) | 40.2 (0.3) | 30.9 (0.5) | 30.8 (0.8) | 43.5 (0.6) | 42.4 (0.4) |
| **FALSE-VFL-I** | 75.7 (0.3) | **73.8** (0.3) | 64.3 (0.7) | **61.4** (0.6) | 61.6 (0.3) | **65.4** (0.9) | 65.5 (0.8) |
| **FALSE-VFL-II** | 76.3 (0.3) | 73.6 (0.7) | **64.6** (0.5) | 61.3 (0.4) | **61.8** (0.4) | 65.3 (0.7) | **66.5** (0.9) |
| | | | | HAPT | | | |
| Vanilla VFL | 79.7 (0.8)$^*$ | 70.4 (1.1)$^*$ | 53.6 (1.7)$^*$ | 57.8 (1.2)$^*$ | 54.7 (1.4)$^*$ | 47.9 (1.8) | 42.7 (4.1) |
| LASER-VFL | 53.1 (11.2) | 54.0 (8.9) | 48.9 (5.8) | 51.0 (5.9) | 49.8 (5.9) | 48.3 (4.9) | 48.1 (6.0) |
| PlugVFL | 84.3 (0.2) | 72.0 (1.3) | 51.4 (1.8) | 58.0 (1.4) | 53.2 (1.5) | 48.1 (2.6)$^*$ | 48.3 (5.4)$^*$ |
| FedHSSL | 78.9 (0.4) | 72.9 (0.5) | 63.1 (0.5) | 63.7 (0.4) | 60.8 (0.6) | 58.6 (0.5) | 54.9 (0.5) |
| **FALSE-VFL-I** | 84.7 (0.1) | 81.2 (0.7) | **71.3** (0.4) | **76.0** (0.1) | **74.1** (0.5) | 67.1 (0.7) | 67.5 (0.6) |
| **FALSE-VFL-II** | **85.1** (0.1) | **81.7** (0.4) | **71.3** (0.1) | 75.3 (0.3) | 73.3 (0.3) | **68.0** (0.6) | **68.1** (0.1) |
| | | | | FashionMNIST | | | |
| Vanilla VFL | 66.6 (2.3) | 58.5 (0.9) | 42.8 (1.9) | 41.5 (1.4) | 41.9 (1.4) | 42.0 (4.0)$^*$ | 41.7 (3.0)$^*$ |
| LASER-VFL | 73.5 (1.2) | 70.8 (0.9) | 64.1 (0.6) | 63.7 (0.7) | 64.8 (1.0) | 63.3 (0.6) | 61.8 (1.5) |
| PlugVFL | 76.0 (0.4) | 72.5 (0.6) | 64.6 (0.8) | 65.0 (0.8) | 65.4 (0.8) | 62.1 (1.4) | 55.8 (2.1) |
| FedHSSL | 71.8 (1.6) | 67.1 (1.6) | 55.5 (1.7) | 54.4 (2.1) | 56.3 (1.8) | 53.6 (2.0) | 50.7 (2.5) |
| **FALSE-VFL-I** | **80.6** (0.1) | **78.1** (0.3) | **67.9** (0.2) | **65.4** (0.3) | **67.5** (0.3) | **66.3** (0.2) | 63.3 (0.3) |
| **FALSE-VFL-II** | 80.5 (0.2) | 77.7 (0.2) | 67.6 (0.4) | 64.7 (0.3) | 67.2 (0.1) | 65.9 (0.3) | **63.4** (0.2) |
| | | | | ModelNet10 | | | |
| Vanilla VFL | 83.9 (0.8) | 76.4 (1.3) | 56.4 (1.7) | 42.2 (2.1) | 39.5 (2.0) | 58.6 (1.4) | 60.5 (1.3) |
| LASER-VFL | 76.3 (2.8) | 73.2 (3.3) | 65.2 (4.9) | 55.5 (4.2) | 54.6 (4.4) | 62.9 (4.7) | 64.6 (5.1) |
| PlugVFL | 84.6 (0.5) | 71.8 (2.1) | 51.2 (3.1) | 45.5 (3.2) | 40.5 (3.6) | 47.8 (3.4) | 45.8 (3.3) |
| FedHSSL | 84.1 (0.8) | 81.9 (0.9) | 70.9 (2.7) | 56.3 (6.6) | 53.8 (6.6) | 71.5 (2.2) | 70.6 (2.4) |
| **FALSE-VFL-I** | 86.8 (0.3) | 85.2 (0.4) | **79.3** (0.4) | **72.1** (0.9) | 70.1 (0.8) | 77.5 (0.7) | 78.3 (0.5) |
| **FALSE-VFL-II** | **86.9** (0.5) | **85.7** (0.1) | 78.9 (0.5) | 71.9 (0.6) | **70.6** (1.0) | **78.7** (0.7) | **79.2** (0.4) |

Table 16: Mean accuracy (%) and standard deviation (in parentheses) over five independent runs for six VFL methods trained under MAR 2 conditions and evaluated on seven test patterns. Boldface highlights the best result in each column. An asterisk (*) marks accuracy computed with labeled data only, as including unlabeled data led to lower accuracy.

| Test Data | MCAR 0 | MCAR 2 | MCAR 5 | MAR 1 | MAR 2 | MNAR 7 | MNAR 9 |
|---|---|---|---|---|---|---|---|
| Isolet | | | | | | | |
| Vanilla VFL | 63.8 (1.1) | 54.9 (0.6) | 35.7 (0.8)* | 26.1 (1.0)* | 26.9 (1.8)* | 37.7 (1.2)* | 38.1 (0.7)* |
| LASER-VFL | 55.3 (6.3) | 50.1 (4.9) | 39.6 (2.6) | 31.6 (2.3) | 32.7 (2.1) | 39.5 (3.3) | 39.7 (2.7) |
| PlugVFL | **78.4** (1.0) | 68.5 (1.0) | 46.7 (0.9) | 35.4 (1.5) | 35.7 (1.0) | 46.4 (1.2) | 46.6 (1.3) |
| FedHSSL | 65.2 (0.6) | 58.3 (0.6) | 40.3 (0.3) | 30.4 (0.3) | 30.4 (0.5) | 43.2 (0.3) | 41.7 (0.7) |
| **FALSE-VFL-I** | 76.7 (0.3) | 74.1 (0.3) | 65.1 (0.7) | **62.1** (0.8) | **62.1** (0.7) | 65.5 (0.8) | 64.8 (0.8) |
| **FALSE-VFL-II** | 78.0 (0.2) | **75.4** (0.2) | **65.4** (0.2) | 60.8 (0.4) | 61.6 (0.6) | **66.0** (1.0) | **66.3** (0.5) |
| HAPT | | | | | | | |
| Vanilla VFL | 81.2 (0.8)* | 72.1 (0.9)* | 55.7 (2.1)* | 59.4 (1.8)* | 56.9 (1.5)* | 51.2 (1.4)* | 46.6 (4.5)* |
| LASER-VFL | 46.9 (12.9) | 49.0 (10.6) | 46.2 (5.1) | 47.9 (6.7) | 46.2 (5.8) | 45.1 (7.2) | 43.6 (11.2) |
| PlugVFL | 84.6 (0.5) | 72.1 (0.6) | 51.1 (0.9) | 57.5 (1.4) | 52.7 (1.2) | 48.5 (3.0) | 48.1 (5.8)* |
| FedHSSL | 79.2 (0.3) | 74.3 (0.3) | 64.6 (0.2) | 65.5 (0.4) | 62.7 (0.6) | 59.4 (0.6) | 55.9 (0.3)* |
| **FALSE-VFL-I** | **86.2** (0.2) | **81.8** (0.2) | **69.6** (0.3) | **74.0** (0.3) | **73.4** (0.6) | **68.0** (0.4) | **68.7** (0.4) |
| **FALSE-VFL-II** | 85.1 (0.3) | 80.5 (0.2) | 69.1 (0.3) | 73.7 (0.6) | 72.1 (0.3) | 66.7 (0.3) | 67.6 (0.4) |
| FashionMNIST | | | | | | | |
| Vanilla VFL | 72.9 (0.4) | 67.6 (0.7) | 53.4 (2.0) | 51.0 (2.5) | 52.2 (2.2) | 49.8 (2.0) | 45.2 (1.4) |
| LASER-VFL | 73.4 (2.5) | 70.9 (2.3) | 64.2 (1.9) | 63.5 (1.6) | 64.4 (1.7) | 63.3 (2.7) | 61.3 (3.6) |
| PlugVFL | 76.8 (0.8) | 73.3 (0.9) | 64.8 (0.7) | 65.0 (0.8) | 65.4 (0.8) | 62.5 (0.9) | 56.8 (0.8) |
| FedHSSL | 71.8 (0.5) | 68.5 (0.8) | 58.6 (2.1) | 56.2 (2.8) | 58.5 (2.6) | 56.1 (2.4) | 51.6 (3.1) |
| **FALSE-VFL-I** | **80.7** (0.2) | **77.9** (0.1) | **67.5** (0.1) | 65.2 (0.4) | **67.6** (0.4) | **66.7** (0.2) | **63.2** (0.3) |
| **FALSE-VFL-II** | 80.4 (0.1) | 77.2 (0.2) | 67.3 (0.4) | **65.3** (0.3) | 67.1 (0.4) | 65.6 (0.4) | 62.4 (0.3) |
| ModelNet10 | | | | | | | |
| Vanilla VFL | 85.3 (0.2) | 76.5 (0.6) | 55.2 (1.2) | 42.3 (0.9) | 39.1 (1.1) | 55.7 (1.3) | 57.4 (1.7) |
| LASER-VFL | 56.6 (9.4)* | 54.3 (7.6)* | 48.2 (6.9) | 42.2 (6.2) | 40.2 (6.0) | 45.2 (7.1) | 42.2 (7.8)* |
| PlugVFL | 84.2 (0.4) | 71.3 (1.7) | 51.0 (2.9) | 42.4 (1.9) | 37.2 (1.8) | 45.8 (2.8) | 44.2 (1.8)* |
| FedHSSL | 85.3 (0.4) | 81.8 (0.9) | 67.8 (3.3) | 53.7 (4.9) | 51.0 (5.4) | 66.1 (4.2) | 64.5 (4.9) |
| **FALSE-VFL-I** | 86.3 (0.5) | 84.6 (0.6) | **78.6** (0.4) | **71.3** (0.5) | **69.7** (0.3) | **77.8** (0.6) | **79.1** (0.7) |
| **FALSE-VFL-II** | **86.6** (0.2) | **85.2** (0.2) | 78.2 (0.5) | 70.9 (0.6) | **69.7** (0.5) | 77.7 (0.6) | **79.1** (0.4) |

Table 17: Mean accuracy (%) and standard deviation (in parentheses) over five independent runs for six VFL methods trained under MNAR 7 conditions and evaluated on seven test patterns.

| Test Data | MCAR 0 | MCAR 2 | MCAR 5 | MAR 1 | MAR 2 | MNAR 7 | MNAR 9 |
|---|---|---|---|---|---|---|---|
| Isolet | | | | | | | |
| Vanilla VFL | 61.8 (1.7) | 52.0 (1.2) | 31.2 (1.9) | 20.2 (3.0)* | 20.6 (1.6) | 31.7 (2.1)* | 32.1 (1.7)* |
| LASER-VFL | 61.7 (5.7) | 56.3 (4.1) | 44.6 (2.3) | 38.7 (2.6) | 39.5 (2.7) | 46.3 (3.0) | 45.6 (3.1) |
| PlugVFL | 81.1 (0.4) | 70.5 (1.3) | 47.7 (2.5)* | 35.5 (2.6)* | 36.9 (3.1)* | 49.4 (1.4)* | 49.4 (1.0) |
| FedHSSL | 65.6 (0.7) | 58.8 (0.7) | 40.9 (1.1) | 30.9 (0.7) | 31.3 (1.0) | 43.5 (0.7) | 42.3 (0.7) |
| **FALSE-VFL-I** | **83.6** (0.5) | **81.2** (0.5) | **69.7** (0.6) | **62.3** (1.0) | **64.4** (1.2) | 70.6 (0.5) | **72.1** (0.5) |
| **FALSE-VFL-II** | 83.3 (0.4) | 80.9 (0.5) | **69.7** (0.4) | 62.1 (0.7) | 63.6 (1.3) | **71.0** (0.5) | 71.9 (0.6) |
| HAPT | | | | | | | |
| Vanilla VFL | 81.7 (0.6) | 72.5 (1.5)* | 55.9 (1.3)* | 59.3 (1.1)* | 56.6 (1.3)* | 50.8 (2.6)* | 44.1 (1.3) |
| LASER-VFL | 61.9 (13.8) | 58.1 (9.3) | 48.9 (3.5) | 54.2 (5.9) | 50.8 (4.8) | 46.5 (3.4) | 49.8 (3.3) |
| PlugVFL | **85.6** (0.3) | 71.6 (0.8) | 50.6 (1.7)* | 58.1 (1.1) | 52.9 (1.4)* | 46.7 (0.9) | 43.9 (0.7) |
| FedHSSL | 80.9 (0.1) | 76.3 (0.4) | 65.8 (0.4) | 66.4 (0.6) | 63.8 (0.5) | 61.6 (0.3) | 58.0 (0.3) |
| **FALSE-VFL-I** | 85.2 (0.3) | 81.7 (0.3) | 72.8 (0.4) | **73.8** (0.6) | 72.4 (0.5) | 71.3 (0.3) | **72.7** (0.4) |
| **FALSE-VFL-II** | 84.9 (0.2) | **81.9** (0.6) | **73.6** (0.4) | 73.6 (0.3) | **73.0** (0.5) | **71.4** (0.6) | **72.7** (0.6) |
| FashionMNIST | | | | | | | |
| Vanilla VFL | 74.1 (0.1) | 68.8 (0.8) | 55.8 (1.8) | 53.2 (2.3) | 54.8 (2.5) | 53.3 (1.8) | 48.6 (1.3) |
| LASER-VFL | 71.0 (4.9) | 67.6 (4.8) | 59.8 (4.5) | 60.9 (4.8) | 60.6 (4.6) | 58.2 (4.8) | 56.5 (5.5) |
| PlugVFL | 76.0 (0.6) | 72.1 (0.7) | 62.9 (1.1) | 63.6 (1.1) | 63.6 (1.0) | 60.9 (0.9) | 56.5 (1.7) |
| FedHSSL | 72.9 (0.4) | 69.5 (0.6) | 59.1 (2.5) | 57.0 (3.5) | 59.6 (2.9) | 56.5 (2.4) | 51.9 (2.3) |
| **FALSE-VFL-I** | 81.1 (0.2) | 78.9 (0.3) | 70.3 (0.3) | **68.1** (0.4) | **70.1** (0.2) | 70.6 (0.3) | 69.7 (0.2) |
| **FALSE-VFL-II** | 81.1 (0.2) | 78.9 (0.1) | **70.4** (0.3) | 67.5 (0.2) | 69.9 (0.3) | 70.6 (0.2) | 69.7 (0.3) |
| ModelNet10 | | | | | | | |
| Vanilla VFL | 84.5 (0.5) | 77.3 (1.0) | 55.8 (1.4) | 50.5 (1.7) | 44.9 (1.9) | 54.4 (2.1) | 53.1 (3.3) |
| LASER-VFL | 79.4 (1.3)* | 73.6 (2.1)* | 63.8 (3.4) | 59.9 (4.6) | 57.1 (4.6) | 63.8 (4.2) | 67.1 (2.5)* |
| PlugVFL | 84.5 (0.5) | 75.0 (1.6) | 56.0 (3.7) | 51.2 (3.7) | 45.5 (4.2) | 53.9 (4.3) | 53.9 (4.9) |
| FedHSSL | 86.6 (0.7) | 83.3 (0.7) | 68.2 (2.8) | 58.7 (2.1) | 55.3 (3.1) | 64.5 (3.1) | 61.0 (3.1) |
| **FALSE-VFL-I** | 87.2 (0.2) | 85.1 (0.4) | 80.9 (0.4) | 74.6 (0.7) | 73.5 (0.3) | 80.5 (0.3) | 82.0 (0.4) |
| **FALSE-VFL-II** | **87.9** (0.4) | **86.0** (0.4) | **81.9** (0.4) | **75.4** (1.0) | **74.4** (0.6) | **81.3** (0.3) | **82.8** (0.4) |

Table 18: Mean accuracy (%) and standard deviation (in parentheses) over five independent runs for six VFL methods trained under MNAR 9 conditions and evaluated on seven test patterns. Boldface highlights the best result in each column. An asterisk ($^*$) marks accuracy computed with labeled data only, as including unlabeled data led to lower accuracy.

| Test Data | MCAR 0 | MCAR 2 | MCAR 5 | MAR 1 | MAR 2 | MNAR 7 | MNAR 9 |
|---|---|---|---|---|---|---|---|
| Isolet | | | | | | | |
| Vanilla VFL | 63.2 (0.9) | 53.4 (1.0)$^*$ | 34.5 (1.4)$^*$ | 24.2 (1.8)$^*$ | 24.0 (2.1)$^*$ | 36.3 (1.5)$^*$ | 36.2 (1.2)$^*$ |
| LASER-VFL | 51.8 (6.9) | 48.4 (4.8) | 40.3 (3.1) | 35.0 (1.5) | 36.2 (2.0) | 42.6 (2.7) | 43.5 (3.3) |
| PlugVFL | 82.9 (0.2) | 72.3 (0.8) | 48.1 (1.2) | 35.4 (1.1) | 35.1 (1.4) | 49.7 (1.3) | 51.4 (1.0) |
| FedHSSL | 65.5 (0.2) | 58.5 (0.7) | 40.0 (0.5) | 29.8 (0.7) | 29.2 (1.0) | 43.2 (0.3) | 41.6 (0.4) |
| **FALSE-VFL-I** | **85.6** (0.2) | 81.9 (0.2) | 69.3 (0.4) | **61.0** (0.4) | **63.7** (0.1) | **72.7** (0.5) | 74.9 (0.3) |
| **FALSE-VFL-II** | 85.2 (0.2) | **82.0** (0.3) | **69.6** (0.5) | **61.0** (0.5) | 63.6 (0.6) | 72.5 (0.4) | **75.1** (0.6) |
| HAPT | | | | | | | |
| Vanilla VFL | 85.4 (0.6)$^*$ | 75.4 (1.1)$^*$ | 56.0 (1.0)$^*$ | 60.8 (1.3)$^*$ | 58.1 (0.9)$^*$ | 49.7 (2.2)$^*$ | 44.2 (1.9)$^*$ |
| LASER-VFL | 59.4 (6.0)$^*$ | 56.8 (6.0)$^*$ | 53.0 (4.9)$^*$ | 57.7 (4.7)$^*$ | 55.8 (4.4)$^*$ | 52.3 (4.8)$^*$ | 55.9 (5.9)$^*$ |
| PlugVFL | **86.5** (0.7)$^*$ | 73.9 (1.1)$^*$ | 53.1 (1.2) | 59.9 (0.7) | 54.7 (0.7) | 49.3 (1.1) | 46.6 (0.8) |
| FedHSSL | 86.4 (0.3) | **81.4** (0.6) | 68.6 (0.7) | **70.6** (0.6) | 68.1 (0.8) | 62.5 (0.6) | 52.9 (0.8) |
| **FALSE-VFL-I** | 83.0 (0.3) | 78.6 (0.7) | 66.9 (0.4) | 67.4 (0.6) | 65.6 (0.2) | 68.1 (0.6) | 72.3 (0.6) |
| **FALSE-VFL-II** | 84.5 (0.5) | 80.2 (0.3) | **69.2** (0.4) | 69.1 (0.3) | **68.3** (0.5) | **69.5** (0.3) | **73.2** (0.5) |
| FashionMNIST | | | | | | | |
| Vanilla VFL | 74.4 (0.4) | 68.7 (0.6) | 52.3 (2.1) | 48.7 (2.2) | 50.5 (2.4) | 50.0 (2.1) | 47.4 (1.7) |
| LASER-VFL | 62.8 (2.9)$^*$ | 59.3 (2.4)$^*$ | 51.1 (1.9)$^*$ | 51.5 (2.5) | 51.9 (2.6) | 50.4 (2.2) | 49.5 (2.5) |
| PlugVFL | 74.9 (0.4) | 70.4 (0.7) | 60.1 (1.9) | 60.4 (1.3) | 60.6 (1.6) | 58.8 (1.3) | 54.8 (1.1) |
| FedHSSL | 74.2 (0.5) | 71.1 (1.0) | 60.3 (3.0) | 57.6 (4.2) | 60.1 (3.7) | 57.8 (2.9) | 53.6 (2.7) |
| **FALSE-VFL-I** | 79.2 (0.2) | 76.3 (0.1) | 65.2 (0.1) | **60.8** (0.4) | **63.9** (0.4) | **68.1** (0.3) | **70.2** (0.2) |
| **FALSE-VFL-II** | **79.6** (0.2) | **76.6** (0.4) | **65.3** (0.4) | 60.2 (0.5) | 63.2 (0.5) | 68.0 (0.2) | 70.1 (0.2) |
| ModelNet10 | | | | | | | |
| Vanilla VFL | 84.5 (0.7) | 76.7 (0.5) | 54.3 (1.8) | 48.6 (1.4) | 41.2 (1.0) | 50.0 (1.3) | 45.1 (3.0)$^*$ |
| LASER-VFL | 78.6 (4.9) | 74.1 (6.0) | 63.8 (6.6) | 60.1 (4.9) | 55.5 (4.6) | 62.9 (7.0) | 65.8 (8.8) |
| PlugVFL | 85.8 (0.6) | 74.0 (0.9) | 54.4 (2.9) | 49.6 (4.8)$^*$ | 44.1 (5.1)$^*$ | 52.7 (3.6) | 53.4 (3.6) |
| FedHSSL | 85.9 (0.4) | 83.7 (0.7) | 73.6 (1.3) | 63.8 (2.5) | 60.8 (3.2) | 70.1 (2.0) | 64.9 (2.4) |
| **FALSE-VFL-I** | **87.1** (0.4) | **85.0** (0.3) | 77.4 (0.4) | **72.1** (0.8) | 70.5 (0.4) | 78.2 (0.4) | 81.3 (0.6) |
| **FALSE-VFL-II** | 86.9 (0.2) | 84.9 (0.2) | **77.6** (0.4) | 71.5 (0.7) | **71.3** (0.9) | **79.7** (0.3) | **81.5** (0.2) |

Table 19: Relative robustness of each method to severe missingness. Each entry is the ratio (in %) of the mean accuracy obtained when training under MCAR 5 to that obtained under MCAR 2, evaluated on seven test patterns. A ratio above 100% means that training with the harsher MCAR 5 mechanism yields higher accuracy than with MCAR 2. Boldface highlights the best ratio in each column.

| Test Data | MCAR 0 | MCAR 2 | MCAR 5 | MAR 1 | MAR 2 | MNAR 7 | MNAR 9 |
|---|---|---|---|---|---|---|---|
| Isolet | | | | | | | |
| Vanilla VFL | 81.6 | 81.7 | 81.7 | 75.3 | 77.8 | 79.4 | 78.3 |
| LASER-VFL | 88.0 | 88.1 | 88.4 | 88.4 | 88.5 | 86.5 | 89.4 |
| PlugVFL | **96.7** | 94.2 | 95.6 | 99.2 | 100.4 | 96.6 | 99.2 |
| FedHSSL | 86.7 | 85.9 | 86.1 | 89.7 | 93.0 | 88.4 | 85.5 |
| **FALSE-VFL-I** | 96.6 | **97.1** | **106.2** | **116.1** | **112.8** | **106.2** | **108.2** |
| **FALSE-VFL-II** | 95.3 | 95.8 | 103.5 | 114.6 | 111.2 | 103.0 | 107.6 |
| HAPT | | | | | | | |
| Vanilla VFL | 92.5 | 92.4 | 97.1 | 93.9 | 95.0 | 100.6 | 96.2 |
| LASER-VFL | **127.2** | **115.6** | 106.1 | 103.8 | 101.8 | **114.0** | **124.7** |
| PlugVFL | 97.7 | 95.8 | 94.0 | 95.8 | 93.7 | 92.7 | 93.0 |
| FedHSSL | 92.5 | 90.9 | 91.5 | 89.9 | 90.1 | 89.9 | 94.6 |
| **FALSE-VFL-I** | 97.2 | 97.2 | 101.4 | 103.1 | 101.7 | 101.9 | 103.9 |
| **FALSE-VFL-II** | 97.9 | 100.4 | **107.4** | **106.3** | **106.3** | 104.2 | 102.7 |
| FashionMNIST | | | | | | | |
| Vanilla VFL | 95.8 | 95.7 | 95.2 | 95.3 | 96.2 | 94.8 | 96.5 |
| LASER-VFL | 95.4 | 95.4 | 96.0 | 96.2 | 95.6 | 95.4 | 94.2 |
| PlugVFL | 98.3 | 97.8 | 97.6 | 99.3 | 98.9 | 95.3 | 93.8 |
| FedHSSL | 92.1 | 91.0 | 89.1 | 90.1 | 90.0 | 87.4 | 86.2 |
| **FALSE-VFL-I** | 97.6 | 98.6 | 103.6 | 103.9 | 103.1 | 103.2 | 101.4 |
| **FALSE-VFL-II** | **98.4** | **99.3** | **104.3** | **105.2** | **104.1** | **103.5** | **101.8** |
| ModelNet10 | | | | | | | |
| Vanilla VFL | 98.9 | 96.6 | 96.9 | 93.9 | 94.2 | 95.4 | 93.9 |
| LASER-VFL | 100.3 | 97.5 | 95.4 | 100.8 | 100.8 | 96.9 | 98.1 |
| PlugVFL | 98.4 | 97.8 | 95.8 | 95.4 | 94.7 | 94.9 | 92.3 |
| FedHSSL | 97.7 | 97.0 | 94.5 | 90.5 | 88.5 | 96.0 | 100.2 |
| **FALSE-VFL-I** | 99.7 | 100.2 | **103.1** | 105.2 | **108.3** | **103.6** | **102.9** |
| **FALSE-VFL-II** | **100.4** | **101.2** | 102.2 | **106.4** | 106.9 | 103.2 | 101.3 |

## F LIMITATIONS AND FUTURE WORK

In FALSE-VFL-II, the mask distribution is modeled as $p_\psi(\boldsymbol{m}|\boldsymbol{x}) = \prod_{k \in [K]} p_{\psi^k}(m^k|\boldsymbol{x}^k)$. This formulation supports the MNAR mechanism, where missingness depends on both observed and unobserved values. However, under this model, each feature vector $\boldsymbol{x}^k$ can only influence the missingness of its own entry $m^k$, and not that of other parties, i.e., $m^l$ for $l \neq k$. In this sense, the current model cannot capture inter-party dependencies in the MNAR mechanism.

Extending our approach to model such inter-party dependencies would be a natural next step. However, doing so poses a significant challenge, as direct sharing of $\boldsymbol{x}^k$ across parties is generally prohibited due to privacy constraints. We leave the development of such models that account for inter-party MNAR dependencies while preserving privacy as an important direction for future work.

## G LLM USAGE

To improve clarity, parts of text were refined with assistance from a large language model; all content was reviewed and verified by the authors.

