# OpenReview forum: "Deep Latent Variable Model based Vertical Federated Learning with Flexible Alignment and Labeling Scenarios"
_ICLR.cc/2026/Conference — ICLR 2026 Poster_

### Official Review · Reviewer_pEAi · 2025-10-20

**Soundness:** 3
**Presentation:** 3
**Contribution:** 3
**Rating:** 6
**Confidence:** 3

**Summary:**

The paper proposes FALSE-VFL, a unified framework for vertical federated learning (VFL) that frames misalignment across parties as a missing-data problem and applies deep latent variable models (DLVMs) to handle arbitrary alignment and labeling scenarios. FALSE-VFL has two variants: FALSE-VFL-I (assumes MAR) and FALSE-VFL-II (models masks and supports MNAR). The method uses a two-stage procedure: (i) pretrain a generative DLVM by maximizing a marginal likelihood over observed features (to exploit unlabeled data), then (ii) freeze the generative parameters and train the discriminative component to maximize conditional likelihood. The paper presents experiments on four benchmarks (Isolet, HAPT, FashionMNIST, ModelNet10) under many training/test missingness regimes (168 configurations total) and reports that FALSE-VFL outperforms baselines in the vast majority of settings, with an average advantage on the order reported in the paper.

**Strengths:**

- The paper reframes VFL alignment problems explicitly as missing-data mechanisms (MCAR/MAR/MNAR) and integrates modern DLVM approaches (IWAE-style training and importance-weighted estimates) into a privacy-conscious VFL pipeline. This synthesis (DLVM + VFL missingness viewpoint + two-stage optimization) is novel in the VFL literature.
- The model design is technically coherent: party-wise encoders/decoders, an aggregated variational posterior with precision averaging, and a label-side discriminator make sense and are explained (mean/precision aggregation rationale). The theoretical properties of the importance-weighted bounds and monotonicity are stated and proved in the appendix.
- The experimental sweep is broad: four datasets, two network families (tabular and image), six training missingness regimes, and seven test regimes, multiple baselines, and ablations (missing rates, number of parties, heterogeneity), altogether 168 configurations reported and per-setting tables provided. Results show consistent wins for FALSE-VFL across nearly all configurations. This breadth strengthens claims of robustness to alignment/label scarcity.

**Weaknesses:**

- FALSE-VFL is claimed as a privacy-preserving VFL method, but the paper seems to lacking a concrete description and analysis of what is communicated at pretraining, training, and inference (e.g., are latent parameters, encoder outputs, importance weights, or reconstructions exchanged?); what information leakages are possible; and whether common VFL privacy protections (secure aggregation, HE, secure two-party protocols) are compatible with the sampling / IWAE machinery used.
- The approach uses importance-weighted sampling and a DLVM pretraining stage, which, as reported, can be expensive (ModelNet10 pretraining = 860 minutes on RTX3090; Table 11). This raises questions about practicality for large real-world VFL deployments with many parties, high-dimensional data, or limited compute.
- The paper misses several important ablation studies: sensitivity to the amount of labeled data (why 500/1000 chosen?), sensitivity to latent dimension, effect of κ (IWAE samples) on final prediction, and when/why FALSE-VFL-I (MAR) sometimes outperforms FALSE-VFL-II despite the latter being more expressive.

**Questions:**

- It is better to provide a precise description of what is communicated between parties in each phase (pretraining, training, prediction). Are raw encoder outputs shared, or only aggregated/statistical information?
- It is better to provide (a) FLOPs or time per epoch for pretraining and training per dataset, (b) communication per sample (bytes exchanged) for inference and training, and (c) how κ and the number of parties K affect runtime and communication.
- The paper aggregates party contributions via mean and precision summation. Could you provide an intuition or visualization showing that h captures cross-party semantics, enabling prediction when parties are missing?

---

> ### Author Response · Authors · 2025-11-18
>
> We thank the reviewer for their careful reading and constructive feedback, and we address the identified weaknesses and questions point by point below.
>
> ---
>
> **Weakness 1.** We appreciate the suggestion to clarify the privacy and communication aspects. In our paper, we use “privacy-preserving” in the standard VFL sense: raw features and labels never leave their local parties, and only intermediate representations and gradients are communicated. The current version already specifies what is exchanged in the algorithm description in Appendix D.3 (e.g., the steps where latent representations are sent from passive to active party), but we agree that this information is easy to miss when it only appears in the appendix.
>
> Concretely, the communication pattern in FALSE-VFL is as follows.
>
> 1. Pretraining and supervised training.
>
> > In both FALSE-VFL-I and FALSE-VFL-II, each party keeps its local encoder/decoder parameters and raw features. For a given sample, each party computes the mean and variance parameters of its approximate posterior and sends only this local latent representation to the active party. The active party aggregates these representations into a global latent distribution, computes the loss, and sends back the corresponding gradients to each party. Each passive party receives only gradients for its own local model parameters. In FALSE-VFL-II, there is an additional exchange for entities with missing parties. The active party samples latent variables from the global posterior and sends these samples to the missing parties. Each such party uses the samples with its local decoder and missing indicator to compute missingness probability, and sends back the probability to the active party.
>
> 2. Inference.
>
> > At prediction time, in FALSE-VFL-I, each participating party applies its encoder to its local data and sends the encoder output to the active party which combines them and produces the prediction. In FALSE-VFL-II, the inference phase mirrors the MNAR training phase. For entities with missing parties, the active party again sends sampled latent variables to the missing parties so that they can compute missingness probability, and these parties send back the probability to be used in the importance weights. No reconstructed features are transmitted.
>
> Thus, FALSE-VFL does not introduce additional channels for raw data leakage beyond standard VFL schemes that share intermediate representations. Our IWAE-based training relies on importance weights and multiple latent samples. In both variants, importance weights are computed only at the active party and are never shared, and in FALSE-VFL-II, the only additional messages beyond encoder outputs and gradients are the sampled latent vectors sent from active party to missing parties and the missingness probabilities returned by those parties.
>
> We will make the communication pattern more visible by adding a short subsection in Section 3 that summarizes what is exchanged in each phase. We will also clarify that our framework is orthogonal to cryptographic protections such as secure aggregation and homomorphic encryption. These techniques can be applied to the exchanged encoder statistics and gradients in the same way as in conventional VFL pipelines.
>
> ---
>
> **Weakness 2.** Regarding the concern that the DLVM pretraining stage is expensive, we would like to clarify how the reported numbers should be interpreted. The 860 minutes reported for ModelNet10 on an RTX 3090 come from an implementation in which all six parties were trained sequentially on a single GPU. In a practical VFL deployment where each party typically performs its local computation on its own device, these local computations can be run in parallel; in that case, the local part of the runtime would be reduced by roughly a factor of six. The large absolute time on ModelNet10 is also driven by the high input dimension and the associated model architecture, rather than by FALSE-VFL itself.
>
> Empirically, even the methods that perform local representation learning at each party such as Vanilla VFL, LASER-VFL, and PlugVFL, take on average about 570 minutes on the same ModelNet10 configuration. Moreover, FedHSSL which performs VFL-based unsupervised learning, requires around 1130 minutes. Given that the pretraining stage of FALSE-VFL is a VFL-based unsupervised learning procedure, these numbers suggest that its pretraining cost cannot be considered unusually expensive compared to existing VFL approaches.

---

> ### Author Response · Authors · 2025-11-18
>
> ---
>
> **Weakness 3.** We appreciate the reviewer’s detailed suggestions about additional ablations and clarify why we made the choices in the current version.
>
> **Amount of labeled data.** Our goal is to place all methods in a regime that clearly reflects label scarcity while still allowing stable training and meaningful accuracy comparisons. For the tabular datasets we use 500 labeled samples, and for the image datasets 1000, so that (i) baselines do not completely collapse due to an extremely small labeled set and (ii) the setting is still far from the fully supervised regime where unlabeled data modeling brings little benefit.
>
> **Latent dimension.** Our focus is to compare the learning frameworks rather than to explore architectural hyperparameters. For each dataset we therefore pick a reasonable latent dimension once and use the same value for all methods. Changing the latent dimension mainly scales the overall capacity of the models in a similar way for every method, and it tends to shift all accuracies together instead of changing the relative ordering. Fixing it per dataset lets us isolate the effect of the proposed training framework. For this reason, we do not view an additional latent dimension ablation as essential for supporting our main claims.
>
> **Effect of the number of importance samples $\kappa$.** As proved in the paper, increasing $\kappa$ makes the importance-weighted objective approach the true marginal likelihood, so in principle larger $\kappa$ is always beneficial for the tightness of the bound. In practice, however, we found that using $\kappa=10$ is already sufficient. Further increasing $\kappa$ does not lead to noticeable gains in predictive accuracy, while it increases computation. For this reason we fix $\kappa$ at 10 throughout.
>
> For FALSE-VFL-I, $\kappa$ affects only the computation at the active party and does not change the communication cost. Each party still sends a single encoder output per sample, independent of $\kappa$. For FALSE-VFL-II, $\kappa$ also controls how many latent samples are sent from the active party to missing parties and how many missingness probabilities are returned, so the communication cost for entities with missing parties grows linearly with $\kappa$. However, these extra messages consist only of low-dimensional latent vectors and scalar probabilities rather than raw features, so the additional communication overhead remains modest.
>
> **When/why FALSE-VFL-I (MAR) can outperform FALSE-VFL-II (MNAR).** Conceptually, FALSE-VFL-II is more expressive since it explicitly models the masks and can represent MNAR mechanisms, whereas FALSE-VFL-I assumes MAR. However, if the underlying missingness mechanism is in fact MAR, this additional flexibility is not needed, and the simpler FALSE-VFL-I, which is easier to train and less constrained by mask parameters, can achieve better performance in some settings. The right way to assess the benefit of FALSE-VFL-II is to look at configurations where both the training and test missingness mechanisms are MNAR. In those MNAR-MNAR regimes, our experiments show that FALSE-VFL-II is at least as good as FALSE-VFL-I and often strictly better.
>
> ---
>
> **Question 1.** Please refer to our response to Weakness 1. We will make the communication pattern more visible by adding a short subsection in Section 3 that summarizes what is exchanged in each phase.

---

> ### Author Response · Authors · 2025-11-18
>
> ---
>
> **Question 2.** We agree that providing more information on computational and communication costs would be helpful. The current submission is written with a primary focus on the methodological side of VFL under arbitrary alignment and labeling conditions, so we did not include a detailed systems style analysis of FLOPs or per epoch communication in the main text. Moreover, since all methods in our experiments were implemented and run sequentially on a single GPU, the absolute wall-clock times are largely implementation- and hardware-dependent and should not be over-interpreted as precise measures of algorithmic efficiency. Our goal is not to propose a communication- or computation-optimal system, but to introduce a probabilistic VFL framework that can handle arbitrary alignment and labeling condtions. As discussed in our response to Weakness 2, the pretraining and training times of FALSE-VFL are of the similar order as existing VFL approaches and are not impractically large in practice.
>
> As we noted in the response to Weakness 3, for FALSE-VFL-I, increasing $\kappa$ affects only computation at the active party and does not change communication. For FALSE-VFL-II, a large $\kappa$ increases the number of latent samples that the active party sends to missing parties and the number of missingness probabilities returned by those parties, but these messages consist only of low-dimensional latent vectors and scalar probabilities.
>
> Regarding the number of parties $K$, the per sample communication in FALSE-VFL grows proportionally with the number of participating parties, as in standard VFL. When parties perform their local computations in parallel on their own devices, the wall clock time per training step is mainly determined by the slowest party and by the active party’s aggregation, rather than increasing linearly with $K$. In this sense, FALSE-VFL does not introduce any additional dependence on $K$ beyond what is already inherent in a standard multi-party VFL setup.
>
> ---
>
> **Question 3.** We can clarify the intuition behind our aggregation rule and how it yields a stable cross-party latent representation.
>
> In FALSE-VFL, each party runs its encoder on its own feature block and outputs a Gaussian posterior over the shared latent variable $h$ with mean and variance. We can think of this as each party giving its own “opinion” about where the latent representation of that sample should lie and how confident it is.
>
> We then combine these opinions in a way that is deliberately designed to have the following semantics:
>
> 1. We aggregate the means by taking an average over the participating parties. This makes the location of $h$ essentially invariant to the number of parties that happen to be present for a given sample. If the same underlying sample is observed by different subsets of parties, their local encoders still map it into the same latent space, and the averaged mean keeps the latent location comparable across different subsets.
>
> 2. We aggregate the variances by summing precisions, which makes the combined variance smaller as more parties participate in. Intuitively, more parties mean more raw features and therefore more information about the same sample, so the model should be more certain about $h$. Our aggregation rule enforces exactly this behavior.
>
> This design is what gives $h$ its “cross-party semantics”. The latent space is shared across all parties and the aggregation rule is defined for any subset of parties. When some parties are missing, we simply aggregate over the available ones. The model has been trained on many different subsets during pretraining and supervised training, so the classifier learns to interpret $h$ consistently whether it comes from any number of parties. Missing parties primarily show up as increased uncertainty, not as a completely different notion of $h$.

---

> ### Comment · Reviewer_pEAi · 2025-11-19
>
> Thanks for the detailed response, and I would like to raise my score.

---

> > ### Author Response · Authors · 2025-11-19
> >
> > Thank you very much for taking the time to reconsider our work and for raising the score. We greatly appreciate your constructive and thoughful feedback.

---

### Official Review · Reviewer_wrRH · 2025-10-27

[review text omitted: it was posted to a different submission]

---

> ### Author Response · Authors · 2025-11-13
> **Apparent mismatch with our submission**
>
> Dear Area Chairs,
>
> We would like to flag that this review appears to be about a different manuscript rather than ours. Concretely, it discusses elements that are absent from our paper and inconsistent with our setup:
>
> 1. The review frames our work as DP-FL and refers to DP-SGD, fixed noise budgets, tighter DP guarantees, and reduced Lipschitz constants. Our submission does not study differential privacy, does not use DP-SGD, and makes no DP claims.
>
> 2. It states that we use latent representations to regularize gradient updates, which is not part of our method.
>
> 3. The review lists datasets MNIST/CIFAR-10/FEMNIST; our experiments are on Isolet, HAPT, FashionMNIST, and ModelNet10.
>
> 4. It mentions DP-FL baselines, which we do not include; we compare against VFL baselines.
>
> Given these discrepancies, could you please verify whether Reviewer wrRH’s comments were mistakenly associated with our submission (e.g., a review swap)? If a mismatch is confirmed, we respectfully ask that the out-of-scope portions be disregarded in the evaluation and, if appropriate, that a replacement review be requested.
>
> Sincerely,
>
> The Authors

---

> ### Author Response · Authors · 2025-11-20
>
> We thank the reviewer for their careful reading and constructive feedback, and we address the identified weaknesses point by point below.
>
> ---
>
> **Weakness 1.** We understand the concern that using a DLVM and multi-party posterior communication might introduce substantial overhead, and we would like to clarify how these components behave in our implementation.
>
> The DLVM-based pretraining stage is an unsupervised learning procedure that is necessary to exploit large amounts of unlabeled data. Unsurprisingly, this kind of unsupervised learning is time-consuming, just as it is for other methods that use local or VFL-based unsupervised learning to handle many unlabeled samples. In our experiments, the largest wall-clock time occurs on ModelNet10, where FALSE-VFL pretraining takes 860 minutes on an RTX 3090 as already reported in the paper. However, all methods in our study were implemented and run sequentially on a single GPU. In a practical VFL deployment, each party would typically perform its local computation on its own device, so the local computations can run in parallel across parties; in such a parallel setting, the portion of the runtime that comes from local computations would be reduced to roughly one sixth in the six-party configuration.
>
> Even within our single-GPU implementation, the methods that perform local representation learning at each party such as Vanilla VFL, LASER-VFL, and PlugVFL already require about 570 minutes of pretraining on the same ModelNet10 configuration, and FedHSSL which performs VFL-based unsupervised learning, requires around 1130 minutes. Because these numbers depend strongly on the single-GPU implementation and are not meant as definitive system-level benchmarks, we did not include them in the main paper, but they indicate that FALSE-VFL is not substantially more time-consuming than existing VFL approaches that also rely on unsupervised learning. After pretraining, the parameters of the generative model are frozen and the supervised training stage optimizes only the classifier, so there is essentially no additional DLVM-specific overhead in this second stage.
>
> Regarding multi-party posterior communication, FALSE-VFL follows the same basic communication pattern as the standard VFL methods that share local representations. In FALSE-VFL-I, each party computes its local representation for a sample which consists of the posterior mean and variance over the shared latent variable, and sends it to the active party. The active party uses these local representations to compute a global representation, computes the loss, and sends back the corresponding gradients to each party. In this sense, the bandwidth usage is of the same order as in the existing VFL algorithms where passive parties send local embeddings to the active party and receive gradients in return. The revised manuscript describes the communicated quantities explicitly: Section 3.5 explains the communication pattern of FALSE-VFL-I, and Appendix A.2 explains the additional messages in FALSE-VFL-II.

---

> ### Author Response · Authors · 2025-11-20
>
> ---
>
> **Weakness 2.** While we fully agree that scalability is an important consideration, our primary goal in this work is to study how to exploit arbitrary alignment patterns and unlabeled data in realistic cross-silo VFL scenarios rather than to push to extreme data and party scales. In VFL, each party is typically an organization (e.g., a hospital or a bank), and VFL is typically deployed in cross-silo settings where a limited number of organizations collaborate. This view is consistent with standard characterizations in the literature which note that HFL can be instantiated in both cross-device and cross-silo settings, whereas VFL is formulated for cross-silo collaboration among a limited number of organizations [1]. In such cross-silo FL scenarios, the practically relevant regime is that the number of participating institutions is small, and the main challenge lies in handling heterogeneous feature splits and complex alignment gaps across these parties. Our experiments are designed with this regime in mind: we consider up to 12 parties, which already stresses the multi-party behavior of the proposed method while allowing us to construct a wide range of alignment configurations.
>
> Within this intended regime, our experimental scale is in fact comparable to, or larger than, what is commonly used in the current VFL literature. Many existing VFL works evaluate only two-party or very small multi-party settings. Our experiments cover up to 12 parties which already lies at the upper end of what is typically considered in the existing VFL studies. Regarding dataset dimensionality, most VFL papers rely on small-to-moderate public benchmarks such as UCI tabular datasets and MNIST- or CIFAR-class image datasets, or on subsampled industrial datasets. In these settings, the dimension of the input features typically ranges from on the order of 10 up to at most a few thousand. In our ModelNet10 experiments, each sample is treated as a six-view example consisting of six 32 by 32 images, so the input dimension per sample is 6144 which is already at least as large as the feature dimensionality commonly explored in prior VFL works. Taken together, these choices place our study firmly within the practically relevant cross-silo regime while demonstrating that the proposed framework scales to party counts and input dimensions that are at least as demanding as those used in the existing VFL evaluations.
>
> [1] Yang Liu, et al. Vertical federated learning: Concepts, advances, and challenges. IEEE Transactions on Knowledge and Data Engineering, 2024.
>
> ---
>
> **Weakness 3.** We agree that intermediate representations can leak information in principle, and that understanding how they interact with secure computation is important for practical deployment. In FALSE-VFL, the messages exchanged between parties are restricted to encoder-level statistics and gradients. All of these are exactly the same type as the embeddings and gradients exchanged in the existing VFL systems.
>
> Because of this, FALSE-VFL is directly compatible with the standard cryptographic and secure-computation techniques. Secure aggregation or homomorphic encryption can be applied to these encoder statistics and gradients in exactly the same way as they are applied to latent representations and gradients in the existing VFL pipelines, without modifying the learning objective or the DLVM-based training algorithm. Our contribution is methodological rather than cryptographic. FALSE-VFL does not by itself eliminate representation leakage, but it is designed so that established privacy-enhancing mechanisms can be layered on top of the exchanged encoder outputs and gradients in a modular way, just as in other VFL methods.

---

> ### Author Response · Authors · 2025-11-20
>
> ---
>
> **Weakness 4.** We understand the concern that our MAR and MNAR mechanisms are synthetically constructed and not directly tied to a particular real-world VFL deployment. Our primary reason for using these synthetic missingness patterns is that FALSE-VFL is explicitly designed to handle both MAR and MNAR mechanisms at the modeling level, and we wanted to evaluate this aspect under controlled, theory-consistent conditions. In particular, for real-world datasets, one typically does not know which missingness mechanism actually generated the data, and this mechanism cannot, in general, be uniquely identified from the observed data alone.
>
> From an evaluation perspective, having a known ground-truth missingness mechanism is important. It allows us to check whether FALSE-VFL-I which assumes MAR and FALSE-VFL-II which can represent MNAR behave as expected when the underlying mechanism is truly MAR or truly MNAR. If we were to use only real-world alignment patterns without knowing whether the underlying mechanism is MAR or MNAR, it would be much harder to interpret whether any performance difference comes from modeling the mechanism correctly versus from incidental dataset-specific artifacts.
>
> We fully agree that testing on more realistic, domain-specific alignment patterns (e.g., multi-hospital medical records or cross-institutional financial data) would further strengthen empirical validity. However, such vertically partitioned datasets with clear alignment gaps are extremely difficult to obtain in practice, because medical and industrial data involving multiple organizations are typically subject to strict privacy regulations, contractual restrictions, and de-identification requirements. For this reason, most existing VFL works also rely on synthetic feature splits and missingness patterns on public datasets.
>
> ---
>
> **Flag For Ethics Review.** Thank you for carefully considering the ethical aspects of our work and for flagging the paper for ethics review. To better understand and appropriately address your concerns in future revisions, could you kindly clarify which specific aspects of our submission motivated the ethics flag? Any additional detail you can provide would be very helpful for us to respond more precisely and to ensure that our presentation is aligned with the conference’s ethical standards.

---

> > ### Comment · Reviewer_wrRH · 2025-11-21
> > **Response to the rebuttal**
> >
> > Thank you for your detailed response. I believe the scalability and the privacy issues are addressed. Could you provide an analysis of the additional computational costs of the proposed method? For instance, the complexity analysis?

---

> > > ### Author Response · Authors · 2025-11-24
> > >
> > > Before addressing to the question, we first clarify a minor point in our previous comment. As in FALSE-VFL-II, in FALSE-VFL-I we also need to send the aggregated global representation from the active party to each passive party in order to compute the $p_{\theta_c}(x^{obs}|h)$ via the local decoders. Each passive party then sends back only the scalar probability, so this communication cost can be regarded as negligible.
> > >
> > > ---
> > >
> > > In response to the question on additional computational cost, we provide a coarse complexity comparison between Vanilla VFL and FALSE-VFL on the tabular datasets used in our experiments. Because each method relies on its own architecture tailored to a different objective, even between Vanilla VFL and FALSE-VFL the following expressions should be viewed as high level MLP based approximations rather than an exact unified comparison across baselines. In what follows, each $O(\cdot)$ term refers to the per step complexity including both forward and backward passes. Note that for fairness, every local model in Vanilla VFL is pretrained with SimSiam.
> > >
> > > In Vanilla VFL, each party has a feature extractor, a projector, and a predictor, and the active party additionally has a discriminator. With input dimension $d$, local representation dimension $h$, number of parties $K$, and number of classes $c$, the computational complexity of each component in our implementation is:
> > >
> > > 1. feature extractor (3-layer MLP: $d\rightarrow h\rightarrow h\rightarrow h$): $O(Kdh + 2Kh^2)$,
> > >
> > > 2. projector (3-layer MLP: $h\rightarrow 8h\rightarrow 8h\rightarrow 8h$): $O(136Kh^2)$,
> > >
> > > 3. predictor (2-layer MLP: $8h\rightarrow 8h\rightarrow 8h$): $O(128Kh^2)$,
> > >
> > > 4. discriminator (2-layer MLP: $Kh\rightarrow h\rightarrow c$): $O(Kh^2 + hc)$.
> > >
> > > Thus the total complexity is $O(Kdh +267K h^2 +hc)$.
> > >
> > > In FALSE-VFL, each party has its own local encoder and decoder, and the active party has global encoder and decoder and the discriminator. Let $\kappa$ denote the number of importance samples used in the DLVM-based training. Then the computational complexity of each component is:
> > >
> > > 1. local encoder (2-layer MLP: $d\rightarrow h\rightarrow h$): $O(Kdh + Kh^2)$,
> > >
> > > 2. local decoder (2-layer MLP: $h\rightarrow h\rightarrow d$, evaluated for $\kappa$ samples): $O(\kappa Kdh + \kappa Kh^2)$,
> > >
> > > 3. global encoder and decoder (3-layer MLPs: $h\rightarrow 8h\rightarrow 8h \rightarrow h/2$, evaluated for $\kappa$ samples): $O(152\kappa h^2)$,
> > >
> > > 4. discriminator (2-layer MLP: $h\rightarrow h\rightarrow c$, evaluated on $\kappa$ samples): $O(\kappa h^2 + \kappa hc)$.
> > >
> > > Hence the total complexity is $O((1+\kappa)Kdh + [(1+\kappa)K + 153\kappa] h^2 + \kappa hc)$.
> > >
> > > Qualitatively, FALSE-VFL thus introduces an additional factor that scales linearly in the number of importance samples $\kappa$, but it preserves the same asymptotic dependence on $(d, h, K, c)$ as Vanilla VFL. Also, note that the global encoder and decoder in FALSE-VFL do not depend on $K$, whereas in Vanilla VFL the number of projector and predictor grows linearly in $K$. In our main experiments, we use $K=8$ and $\kappa=10$.

---

> ### Comment · Reviewer_wrRH · 2025-11-24
>
> Thank you for your response. Overall, I believe it is an interesting paper that investigates different settings of misalignment (missing data) in VFL. I am willing to raise my score.

---

> > ### Author Response · Authors · 2025-11-24
> >
> > Thank you very much for taking the time to reconsider our work and for raising the score. We greatly appreciate your constructive and thoughful feedback.

---

### Official Review · Reviewer_JZzY · 2025-10-28

**Soundness:** 3
**Presentation:** 2
**Contribution:** 3
**Rating:** 8
**Confidence:** 4

**Summary:**

This paper proposes FALSE-VFL, a deep latent variable model (DLVM) framework designed for vertical federated learning (VFL). The core idea is to reframe VFL alignment gaps as a blockwise missing data problem. This approach allows the model to handle arbitrary data alignment, leverage abundant unlabeled data, and support multi-party collaboration. The method employs a two-stage optimization: (1) an unsupervised, generative pretraining phase to learn a robust data representation from all samples, and (2) a supervised training phase to learn a predictor from the scarce labeled data . The paper introduces two variants, FALSE-VFL-I for MAR and FALSE-VFL-II for MNAR settings , and demonstrates state-of-the-art performance across 168 experimental configurations

**Strengths:**

S1. The paper tackles a critical and highly practical challenge in VFL. The assumptions that data alignment is imperfect and labels are scarce are far more realistic than in typical VFL literature.

S2. This is a unified framework that addresses multi-party VFL, arbitrary (blockwise) data missingness, and semi-supervised learning. The explicit handling of all three missingness mechanisms (MCAR, MAR, and MNAR) is a significant contribution.

S3. The experimental results are extensive and compelling. The method's dominance in 160 out of 168 configurations, with a large average performance gap over the next-best competitor, provides strong evidence of its effectiveness.

S4. The method shows admirable robustness not only to different types and degrees of missingness but also to an increasing number of parties and data heterogeneity, both of which are common challenges in real-world VFL systems.

**Weaknesses:**

W1. The writing in the methodology section (Section 3) could be significantly improved. The paper is not self-contained and is difficult to follow without prior knowledge of the cited works (e.g., IWAE, Ipsen et al., 2022).
* For instance, the paper states, "so we need some tricks as in IWAE" but fails to explain the intuition behind this "trick" (i.e., importance-weighted sampling as a variational lower bound).
* Similarly, the derivation of the lower bound $\mathcal{L}_{\kappa}$ omits the key appeal to Jensen's Inequality, which is fundamental to understanding why it is a lower bound. While this proof is in the appendix, its absence in the main text makes the core methodology less accessible.

W2. The paper proposes two distinct solutions: FALSE-VFL-I for MAR and FALSE-VFL-II for MNAR . This separation is a practical weakness. In real-world scenarios, the true missingness mechanism is rarely known in advance. A practitioner would not know whether to deploy model I or model II. A more robust and unified framework that could handle (or even infer) the missingness mechanism automatically, without requiring this a priori assumption, would be a significant improvement.

**Minor Comments**

C1. The paper consistently and incorrectly refers to its Appendices as "Sections" in the main text. For example, "described in Section A.2" , "explained in Section B" , and "See Section C for details".

C2. Equation Numbering: Several key equations in the main methodology (Section 3.4) are not numbered. Numbering these equations would make the method much easier to follow and reference.

C3. The meaning of $z_j$ is not explicitly explained in the main paper. The paper would be clearer if it explicitly stated that these are the $j$-th samples (out of $\kappa$) drawn from the approximate posterior distributions.

C4. The meaning of the indices for the missingness mechanisms is not fully explained in the main paper. While MCAR 0/2/5 and MNAR 7/9 are clearly defined in Section 5.1 as referring to probabilities, the crucial distinction between MAR 1 and MAR 2 is not. The main text only states that "Precise formulas are given in Section D.4". For the paper to be self-contained, a brief, one-sentence summary of what "MAR 1" and "MAR 2" represent should be included in Section 5.1.

**Questions:**

See weakness.

---

> ### Author Response · Authors · 2025-11-18
>
> We thank the reviewer for their careful reading and constructive feedback, and we address the identified weaknesses point by point below.
>
> ---
>
> **Weakness 1.** We thank the reviewer for pointing out that Section 3 could be made more self-contained.
>
> In the revised version, we will replace the informal reference to a “trick” with a brief mention of the underlying importance-weighted sampling. We will also make the derivation of our lower bound more explicit in the main text by moving the main intuition and theorem from the appendix into Section 3. This will make the core methodology easier to follow without relying on the appendix.
>
> ---
>
> **Weakness 2.** We agree that it would be desirable to have a single model that does not require the practitioner to decide in advance whether the missingness mechanism is MAR or MNAR.
>
> From a modeling standpoint, parametric mask models that are used for MNAR are defined so that the missingness probability can depend on both observed and missing values. Within such modeling, MAR appears as a special case obtained by constraining the parameters so that the mask no longer depends on the unobserved components. In this sense, MNAR based model can in principle cover both MAR and MNAR regimes, so a practitioner could use the same model even without knowing the true mechanism beforehand.
>
> At the same time, we want to be precise about what FALSE-VFL-II actually models. As discussed in Appendix F, our MNAR variant does not capture all possible inter-party dependencies in the most general MNAR setting, so it is not a completely unrestricted MNAR model. A fully general MNAR mechanism over vertically partitioned features would require a substantially more complex structure, which is beyond the scope of this paper, although we view it as an interesting direction for future work.
>
> The motivation for keeping both FALSE-VFL-I and FALSE-VFL-II is both conceptual and practical. When the underlying mechanism is close to MAR, the simpler FALSE-VFL-I can be easier to train and, in some regimes, can achieve slightly better performance. FALSE-VFL-II is designed to be more flexible when MNAR structure is present. Empirically, our experiments show that both variants perform well across a range of MAR and MNAR regimes. This suggests that in practice both variants are reasonably robust to moderate misspecification of the true mechanism, even though developing a fully unified and mechanism agnostic model remains an important open problem.
>
> ---
>
> **Minor Comments.** We will carefully address all of these points in the revised version. For C2, we would like to clarify whether it is necessary to number equations that are not referenced later in the text, as currently do not refer back to those equations.

---

> > ### Comment · Reviewer_JZzY · 2025-11-25
> >
> > I thank authors for their response. My concern of writing has been resolved. A unified framework for both MAR and MNAR case could be left for authors' future investigation. After reading the response and the revised manuscript, I decide to maintain the current positive rating.

---

> > > ### Author Response · Authors · 2025-11-25
> > >
> > > We sincerely thank the reviewer for carefully reading our rebuttal and revised manuscript and for maintaining your positive rating. We are glad that the concerns about the writing have been resolved. We also agree that developing a unified framework that simultaneously handles both MAR and MNAR is an important and interesting direction, and we appreciate the reviewer for highlighting this avenue for future investigation.

---

### Official Review · Reviewer_nHhw · 2025-10-31

**Soundness:** 2
**Presentation:** 3
**Contribution:** 2
**Rating:** 4
**Confidence:** 4

**Summary:**

This paper introduces FALSE-VFL (Flexible Alignment and Labeling Scenarios Enabled Vertical Federated Learning), a unified probabilistic framework for vertical federated learning (VFL) that can operate under arbitrary combinations of data alignment and labeling conditions. The method models feature, label, and alignment missingness jointly using a Deep Latent Variable Model (DLVM) and formalizes training under MCAR, MAR, and MNAR mechanisms. Experiments on multiple datasets show that FALSE-VFL consistently outperforms existing methods.

**Strengths:**

1. The paper addresses an important limitation of current VFL research, where some methods assume perfect alignment and full labeling. 2. Treating alignment and labeling gaps as missing-data problems and modeling them jointly within a probabilistic latent-variable framework is a conceptually coherent way to generalize VFL beyond idealized settings.
3. The experiments are comprehensive, covering many combinations of training and testing conditions, and they demonstrate robust performance improvements.
4. The paper is clearly written and well-motivated.

**Weaknesses:**

1. The paper's main limitation lies in the originality of its methodological foundation. The proposed DLVM-based framework for modeling unaligned, unlabeled, and partially observed data in VFL largely parallels the broader line of work on robust FL already explored in horizontal settings. Prior studies [R1-R3] have developed principled approaches for robustness under heterogeneous, noisy, or distributionally shifted clients. While these works operate under horizontal data partitioning, they share the same conceptual goal as FALSE-VFL: learning representations and aggregation mechanisms that remain stable in the presence of incomplete or inconsistent local data.

R1: Robust Federated Learning With Noisy and Heterogeneous Clients, CVPR, 2022

R2: Robust Federated Learning: The Case of Affine Distribution Shifts, NeurIPS, 2020

R3: A Systematic Literature Review of Robust Federated Learning: Issues, Solutions, and Future Research Directions, ACM Computing Survey, 2025

2. Compared with these precedents, the present paper’s innovation lies mainly in applying the robustness principle to vertical feature partitions using deep latent-variable modeling, rather than proposing a fundamentally new robustness mechanism. The model treats missing features, labels, and alignment as latent variables and marginalizes them via variational inference. While this probabilistic formulation is elegant, the underlying idea of achieving federated robustness through representation consistency is well established. The paper does not provide theoretical or empirical evidence that the DLVM approach offers advantages over alternative robustness strategies such as adversarial minimax optimization or noise-weighted aggregation.

3. The framework implicitly assumes that some aligned or labeled samples exist to anchor the shared latent space. This reliance makes the method closer to pseudo-matching alignment strategies than to fully unsupervised alignment inference. The empirical evaluation demonstrates broad coverage but remains descriptive; ablation studies isolating the contributions of missingness modeling or pretraining are limited.

**Questions:**

1. The paper’s design philosophy seems to resemble prior robust FL methods. Can you clarify what new robustness mechanism is introduced by FALSE-VFL beyond adopting a deep latent-variable model for the vertical setting?
2. Could the proposed probabilistic modeling of missingness be replaced by other robustness strategies, such as adversarial or confidence-weighted training, and if so, what unique advantage does the DLVM formulation offer?
3. Does the method require a subset of aligned or labeled samples to establish a shared latent representation? If none are available, how would the model avoid degenerate alignment?
4. Can you provide empirical or theoretical evidence that the probabilistic marginalization approach improves robustness relative to prior adversarial or reweighting-based techniques?

---

> ### Author Response · Authors · 2025-11-18
>
> We thank the reviewer for their careful reading and constructive feedback, and we address the identified weaknesses and questions point by point below.
>
> ---
>
> **Weakness 1.** Horizontal FL (HFL) and vertical FL (VFL) address fundamentally different problem domains and therefore require different model designs. In HFL, all parties work in the same input feature space, so each party trains its local model on full feature vectors and the server aggregates the local models, either by averaging their parameters or by aggregating their outputs on public data; there is no need to merge local representations within a single forward pass. In contrast, VFL operates on feature-partitioned data across parties; different parties hold disjoint subsets of a joint feature space, and their encoders must be combined at the representation level to produce a single prediction for each sample. This introduces a notion that does not appear in HFL, namely alignment of samples across parties. For any given record, some parties may have the corresponding feature block and others may not, so records can be fully aligned, partially aligned, or fully unaligned.
>
> Our paper does not aim to transfer a generic robustness idea from HFL. Our main focus is the following VFL-specific question:
>
> *When alignment patterns are arbitrary and many samples are unlabeled, how can we still leverage all available data in a principled way in VFL?*
>
> This leads to a methodological contribution that, to the best of our knowledge, has not been explored in prior VFL works: a unified framework that supports both training and inference under arbitrary alignment and labeling conditions in multi-party VFL. The observed robustness to higher missing rates, larger numbers of parties, and data heterogeneity are then an empirical consequence of our framework as demonstrated in our experiments.
>
> ---
>
> **Weakness 2.** The review characterizes our method as merely instantiating an already established idea of “representation consistency” for federated robustness. We believe this does not accurately reflect the methodological contribution of FALSE-VFL. Our setting is VFL with arbitrary alignment patterns and substantial label scarcity, which is structurally different from the HFL scenarios where most robustness techniques have been developed.
>
> Methodologically, FALSE-VFL is built on a likelihood-based deep latent variable model for vertically partitioned data with arbitrary alignment and a two-stage training scheme that can handle arbitrary alignment and label scarcity as follows:
>
> 1. For handling arbitrary alignment, we interpret an incomplete alignment across parties as a form of blockwise missingness in the feature space and learn a shared latent representation that can be inferred from any subset of parties.
>
> 2. For handling label scarcity, we first maximize the marginal likelihood to learn the generative model and latent space while leveraging unlabeled samples, and then maximize the conditional likelihood using the labeled subset to train the predictor on top of this latent space.
>
> To the best of our knowledge, FALSE-VFL is the first VFL method that combines (i) a DLVM-based treatment of arbitrary alignment as blockwise feature missingness with (ii) a two-stage training to exploit unlabeled data, while supporting arbitrary alignment patterns during both training and inference. This is precisely what allows FALSE-VFL to remain effective under high missing rates and heterogeneous missing rates across parties.
>
> Regarding the comment that we provide no evidence that a DLVM is preferable to adversarial minimax optimization or noise-weighted aggregation, we note that these techniques have been developed for HFL where all parties share the same feature space and therefore cannot be straightforwardly applied to our VFL setting. Our aim is not to claim that DLVMs universally dominate adversarial or reweighting-based methods, but to propose a probabilistic framework tailored to vertically partitioned data with arbitrary alignment. Within this VFL setting, FALSE-VFL consistently outperforms strong VFL baselines under challenging alignment and labeling conditions, which supports the practical value of our framework.

---

> ### Author Response · Authors · 2025-11-18
>
> ---
>
> **Weakness 3.** First, we would like to clarify what FALSE-VFL actually assumes. FALSE-VFL assumes that (i) some samples have labels and that (ii) some samples appear in more than one party, so that multiple parties each hold their own feature block for the same sample. The overlapping samples do not need to be fully aligned across all parties; partially aligned samples are sufficient as long as there is some overlap so that feature information from multiple parties can be jointly used. If no labels exist at all, the problem is no longer a supervised learning task which is the focus of our work. If, in addition, all samples were fully unaligned so that no sample ever appears in more than one party, then there would be no cross-party overlap to exploit and the setting would effectively fall outside the usual notion of VFL.
>
> Second, although the reviewer relates our method to pseudo-matching, FALSE-VFL does not construct pseudo-aligned pairs or artificially expand an anchor set. Unaligned records remain unaligned in our framework; we do not infer synthetic correspondences between parties beyond the alignment that is actually observed. Conceptually, we assume that the given alignment pattern, including partial alignment across parties, is maximal. As stated in the introduction, we focus on inherently unalignable records where the observed alignment across parties cannot be further linked, rather than on potentially alignable yet currently unlinked records that could in principle be recovered by a separate record-linkage procedure. In FALSE-VFL, these alignment gaps are handled by treating the missing feature blocks within a deep latent variable model and marginalizing over the unobserved parts, rather than by enforcing additional matches.
>
> Third, regarding the comment that the empirical evaluation is descriptive, our intention is not to show a single illustrative configuration but to evaluate FALSE-VFL under a wide range of conditions. We already report results under diverse alignment patterns, multiple missing rates, heterogeneous missing rates across parties, and different numbers of parties. In our design, the unsupervised marginal likelihood stage is not an optional pretraining trick but an integral part of the method whose purpose is to make effective use of scarce labels. Removing this stage results in replacing FALSE-VFL with a different algorithm that no longer follows our probabilistic formulation, so we view such an ablation as less informative than the variations in alignment, missing rates, and party configuration that we already provide.
>
> ---
>
> **Question 1.** Please refer to our responses to Weaknesses 1 and 2, which clarify how FALSE-VFL differs conceptually and methodologically from prior robust HFL methods.
>
> ---
>
> **Question 2.** Our proposed probabilistic modeling of missingness cannot simply be replaced by other robustness strategies such as adversarial or confidence-weighted training. These strategies have been developed for HFL algorithms where all parties share the same feature space, and are therefore not directly applicable to VFL setting. In our work, we further focus on VFL with arbitrary alignment and labeling conditions, which makes a straightforward adaptation of these HFL-based methods even less appropriate and outside the scope of this paper.
>
> ---
>
> **Question 3.** As explained in our response to Weakness 3, our method requires (i) some labeled samples and (ii) some samples that are not fully unaligned. First, if there are no labeled samples at all, the problem is no longer a supervised learning task which is the focus of our work. Second, if all samples are fully unaligned, then the problem would fall outside the usual VFL setting. If these two conditions are satisfied, FALSE-VFL does not learn alignment from scratch but instead treats the observed (partial) alignment as given and models the remaining missing feature blocks within the DLVM, so there is no issue of degenerate alignment.
>
> ---
>
> **Question 4.** Please refer again to our responses to Weaknesses 1 and 2. Prior adversarial or reweighting-based techniques have been developed for HFL under very different structural assumptions and are not directly applicable to vertically partitioned data with arbitrary alignment and labeling conditions. As a result, a direct empirical or theoretical comparison within our VFL setting is not well defined and falls outside the scope of this paper. Instead, we provide extensive empirical evidence that our probabilistic approach improves robustness over strong VFL baselines under challenging conditions.

---

> ### Author Response · Authors · 2025-11-26
> **Gentle Reminder**
>
> Dear Reviewer nHhw, this is a gentle reminder in case our earlier response to your comments was missed. We would greatly appreciate it if you could briefly revisit our reply and share any remaining concerns or questions during the discussion period.

---

> > ### Comment · Reviewer_nHhw · 2025-11-26
> >
> > Thanks for the responses. They have addressed my concerns. I will raise my score.

---

> > > ### Author Response · Authors · 2025-11-26
> > >
> > > Thank you very much for taking the time to reconsider our work and for raising the score. We greatly appreciate your constructive and thoughful feedback.

---

### Meta-Review · Area_Chair_LM7N · 2026-01-07

**Summary:**

Reviewers broadly agree that the paper addresses a highly important and realistic problem in vertical federated learning by relaxing idealized assumptions on perfect alignment and abundant labels. Framing alignment gaps, label scarcity, and feature missingness as missing-data mechanisms within a deep latent-variable framework is seen as conceptually coherent, and the extensive experimental evaluation demonstrates strong empirical robustness and consistent performance gains. However, several concerns temper the contribution. Methodologically, the work is viewed as an application of established robustness and latent-variable principles to the VFL setting rather than a fundamentally new robustness mechanism, with limited evidence that the DLVM approach outperforms alternative strategies. Theoretical support is considered incomplete, particularly regarding privacy guarantees and robustness claims, and key assumptions (e.g., partial alignment or labels, known missingness mechanisms) reduce practical generality. Reviewers also note insufficient clarity in the methodological exposition, lack of unified handling of MAR/MNAR scenarios, limited ablations, and unanswered questions about scalability, computational overhead, and concrete privacy leakage risks—especially given non-private pretraining stages. Overall, the paper is seen as empirically strong and well-motivated. Therefore, AC's recommendation is to accept as a poster paper.

**Reviewer Concerns:**

All reviewers' questions were properly addressed during the rebuttal period.

**Reviewer Scores:**

Three reviewers are willing to increase the rating. I expect the final rating to be as follows:
- Reviewer nHhw: 6
- Reviewer JZzY: 8
- Reviewer wrRH: 6
- Reviewer pEAi: 8

---

### Decision · Program_Chairs · 2026-01-26

Accept (Poster)